# CXCL12 and MYC control energy metabolism to support adaptive responses after kidney injury

Toma A. Yakulov [1], Abhijeet P. Todkar [1], Krasimir Slanchev[1,11], Johannes Wiegel [1], Alexandra Bona[1,2,3], Martin Groß[1,2], Alexander Scholz[1], Isabell Hess[1], Anne Wurditsch[1], Florian Grahammer[1,4], Tobias B. Huber[1,4,5], Virginie Lecaudey[2,5,12], Tillmann Bork[1], Jochen Hochrein[6], Melanie Boerries[6,7], Justine Leenders[8], Pascal de Tullio[8], François Jouret [9,10], Albrecht Kramer-Zucker[1] & Gerd Walz [1,5]

Kidney injury is a common complication of severe disease. Here, we report that injuries of the zebrafish embryonal kidney are rapidly repaired by a migratory response in 2-, but not in 1-day-old embryos. Gene expression profiles between these two developmental stages identify *cxcl12a* and *myca* as candidates involved in the repair process. Zebrafish embryos with *cxcl12a*, *cxcr4b*, or *myca* deficiency display repair abnormalities, confirming their role in response to injury. In mice with a kidney-specific knockout, *Cxcl12* and *Myc* gene deletions suppress mitochondrial metabolism and glycolysis, and delay the recovery after ischemia/reperfusion injury. Probing these observations in zebrafish reveal that inhibition of glycolysis slows fast migrating cells and delays the repair after injury, but does not affect the slow cell movements during kidney development. Our findings demonstrate that *Cxcl12* and *Myc* facilitate glycolysis to promote fast migratory responses during development and repair, and potentially also during tumor invasion and metastasis.

[1] Renal Division, University Freiburg Medical Center, Faculty of Medicine, University of Freiburg, Hugstetter Strasse 55, 79106 Freiburg, Germany. [2] Faculty of Biology, University of Freiburg, Schaenzlestrasse 1, 79104 Freiburg, Germany. [3] Spemann Graduate School of Biology and Medicine (SGBM), University of Freiburg, Albertstrasse 19A, 79104 Freiburg, Germany. [4] III. Department of Medicine, University Medical Center Hamburg-Eppendorf, Martinistrasse 52, 20246 Hamburg, Germany. [5] BIOSS Center for Biological Signalling Studies, University of Freiburg, 79104 Freiburg, Germany. [6] Institute of Molecular Medicine and Cell Research, Faculty of Medicine, University of Freiburg, Stefan-Meier Strasse 17, 79104 Freiburg, Germany. [7] German Cancer Consortium (DKTK), Freiburg and German Cancer Research Center (DKFZ), 69120 Heidelberg, Germany. [8] Center for Interdisciplinary Research on Medicines (CIRM), Metabolomics Group, University of Liège, Avenue Hippocrate 15, 4000 Liège, Belgium. [9] Division of Nephrology, University of Liège Hospital, 4000 Liège, Belgium. [10] Groupe Interdisciplinaire de Génoprotéomique Appliquée (GIGA), Cardiovascular Sciences, University of Liège, 4000 Liège, Belgium. [11] Present address: Max-Planck-Institute of Neurobiology, Am Klopferspitz 18, 82152 Martinsried, Germany. [12] Present address: Institute for Cell Biology and Neuroscience, Goethe-University Frankfurt, Max-von-Laue-Strasse 13, 60438 Frankfurt am Main, Germany. These authors contributed equally: Toma A. Yakulov, Abhijeet P. Todkar, Krasimir Slanchev, Johannes Wiegel. Correspondence and requests for materials should be addressed to G.W. (email: gerd.walz@uniklinik-freiburg.de)

Zebrafish are equipped with a fully functional kidney (pronephros) to maintain electrolyte and water homeostasis during embryogenesis[1,2]. One day post fertilization (1 dpf), the zebrafish proximal tubule spans more than half of the pronephros, but is rapidly condensed to the most proximal region within the next 2 days of embryogenesis[3]. This change is caused by collective cell migration that originates in the distal pronephros, and results in formation of a hairpin-like convolution of the proximal pronephric tubules adjacent to the single glomerulus of the zebrafish embryonic kidney by 3 dpf, resembling the convoluted proximal tubules of mammalian nephrons. The cell migration starts at a rate <2 μm/h during the first 24 hours post fertilization (hpf), reaching a maximal speed of 6–8 μm/h between 36 and 48 hpf with a sharp increase from 2 to 6 μm/h at 28.5 hpf[3]. Ablation of tubular epithelial cells in zebrafish embryos is rapidly repaired by a migratory response that occurs independently of cell proliferation, restoring the integrity of the zebrafish pronephros[4].

Collective cell migration is also observed during the development of the posterior lateral line (pLL), a mechano-sensory system found in aquatic vertebrates to detect water movements. Migration of the pLL primordium (pLLP) requires intact Cxcl12a/Cxcr4b signaling[5,6]. Zebrafish Cxcl12a activates Cxcr4b at the leading end of the primordial cell cluster to initiate directed cell migration, while Cxcr7b acts as a decoy receptor for Cxcl12a at the trailing end, creating a local gradient for Cxcl12a[7,8]. Mutation of any of these three components results in defective pLLP development. Mammalian CXCL12/CXCR4 signaling is involved in many cellular programs ranging from directed cell migration and organ development to self-renewal of hematopoietic stem cells and tumor metastasis[9]. The chemokine CXCL12 recognizes the G-protein-coupled receptor CXCR4, which initiates signal transduction by stimulating heterotrimeric G proteins of the $G_i$, $G_q$, and $G_{12/13}$ families, followed by activation of MAPK and phospholipase C pathways[9,10]. CXCL12 promotes cell survival through activation of AKT and mTOR[11,12].

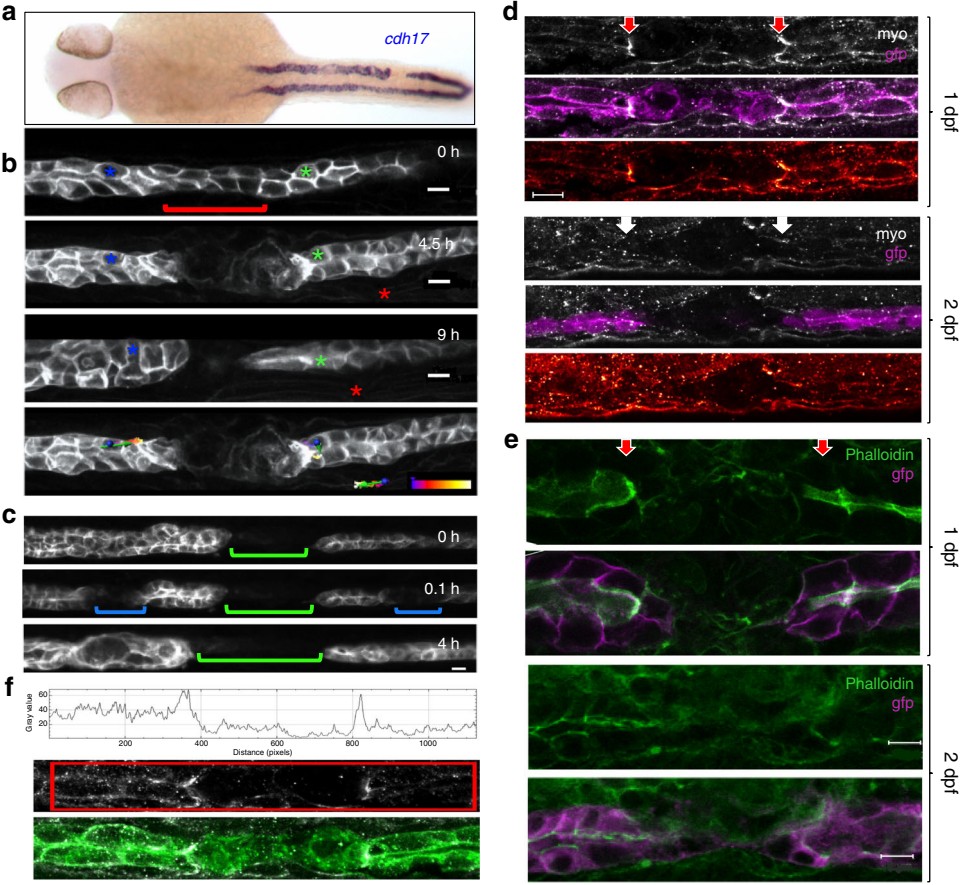

**Fig. 1** Pronephric ducts injured 1 day after fertilization fail to repair. **a** In situ hybridization for the pronephros-specific probe *cadherin17* (*cadh17*) reveals that an injury applied 24 h after fertilization (24 hpf) is not repaired. The in situ hybridization was performed 24 h after injury. **b** Frames taken from a time-lapse movie of *cldn2b:lyn-GFP* transgenic embryo injured 30 hpf. Cells (blue and green asterisks) next to the gap exhibit little net movement. The gap persisted for 9 h after the injury. The red asterisk marks a somite border serving as a landmark. Tracking lines are color coded, ranging from blue to white (0–9 h) (scale bars, 10 μm). **c** Repair occurs in the absence of fluid flow. Frames were taken from a time-lapse movie injured first 1 day post fertilization (dpf) (green bracket), with two subsequent injuries (blue brackets) applied 2 dpf. Despite the lack of fluid flow both secondary injuries (blue brackets) recovered after 4 h. **d** Staining of *cldn2b:lyn-GFP* transgenic zebrafish embryos wounded at 1 or 2 dpf, fixed 30 min post wounding, and stained for phospho-myosin light chain-2 (MyoP). Activated myosin was detected at the edges of the pronephros injury in 1-day old embryos (red arrows), but not in 2-day-old embryos (white arrows). The bottom panel of each image shows the phospho-myosin levels using a color-coded range indicator (scale bars, 100 μm). **e** Staining of *cldn2b:lyn-GFP* transgenic zebrafish embryos wounded 1 or 2 dpf, fixed 4 h after wounding, and stained with phalloidin for actin. Actin staining was detected on the apical surface of cells close to the injury in both 1-day and 2-day old embryos. Actin staining was more prominent in 1-day embryos, and appeared perpendicular to the duct lumen (red arrows) (scale bars, 100 μm). **f** Quantification of the pixel intensity within the marked region of a single representative confocal plane from the MyoP staining (middle panel) of an embryo injured 1 dpf. The histogram depicts two peaks with increased levels of phosphorylated myosin adjacent to the injury. The bottom panel represents the merged image of GFP/MyoP staining

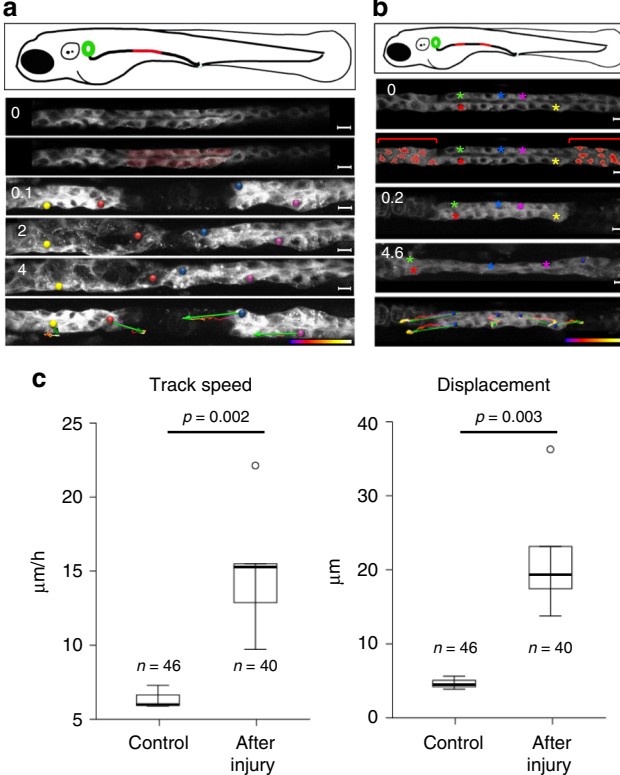

**Fig. 2** Pronephric duct injuries are repaired by migration. **a** Frames from Supplementary Movie 1, showing recovery after ablation of a pronephric duct fragment (ablated cells are marked in red). The free ends of the duct move towards each other, whereby the leading cells extend protrusions and exhibit active cell migration. The bottom panel depicts tracking of individual cells over time. Tracking lines are color coded, ranging from blue to white (0–9 h); the green arrows summarize migration direction and cell displacement (scale bars, 10 µm). **b** Frames from Supplementary Movie 2. The panel depicts a two-sided cell ablation (red-labeled cells), isolating a patch of pronephric cells. The free ends of the isolated duct migrated in opposite direction, stretching the isolated segment. Whereas the cells in the terminal regions exhibited prominent net displacement, the cells in the middle segment remained almost stationary. Tracking lines are color coded, ranging from blue to white (0–9 h) (scale bars, 10 µm). **c** To compare repair response and collective cell migration, 2-day-old Tg(-8.0cldnb:LY-EGFP), TgBAC(cxcr4b:h2b-RFP) transgenic zebrafish embryos were mounted on the side to image pronephric ducts by confocal microscopy. The pronephros was injured with a two-photon laser at 36 h, and track speed (left panel) ($p$ = 0.0023, $t$-test) and track displacement length (right panel) ($p$ = 0.0034, $t$-test) was measured over 2 h. After injury, tubular epithelial cells adjacent to the injury almost tripled their speed. The cell number ($n$) was derived from three control and six injured pronephric tubules

Comparing different stages of zebrafish pronephros development, we identify *cxcl12a* and *myca* as candidate genes involved in the repair process after a laser-induced pronephros injury. Since the cellular behavior of tubular epithelial cells of the zebrafish pronephros strongly resembles pLLP migration, we anticipated similar roles for Cxcl12a and Cxcr4b during pLLP and zebrafish pronephros development. However, the absence of either *cxcl12a* or *cxcr4b* did not affect normal cell migration and development of the zebrafish pronephros. Instead, *cxcl12a/cxcr4b* as well as *myca* were needed for the repair response after an injury to transiently override the posterior-to-anterior collective cell migration in the zebrafish pronephros. Utilizing mice with kidney-specific deletion of *Cxcl12* or *Myc* reveals that both CXCL12 and MYC preserve tubular responsiveness to ischemia

by maintaining mitochondrial homeostasis and supporting the switch from aerobic to anaerobic energy production.

## Results

**Early zebrafish pronephros injuries fail to recover.** Collective posterior-to-anterior cell migration drives pronephros morphogenesis during early zebrafish kidney development[3]. To determine when pronephric cells acquire the capacity to regenerate an injured duct by a migratory response, we ablated parts of the pronephric duct at different time points after fertilization, and followed the repair process. In contrast to the regeneration of injuries introduced after 36 hpf (>36 hpf), embryos injured before 30 hpf failed to re-establish the integrity of the duct (Fig. 1a). The bordering ends of the disrupted duct remained separated, and the length of the gap increased over time (Fig. 1b, c). The lack of regeneration was not attributable to the absence of fluid flow, since pronephric tubules immediately dilated proximal to the injury (Fig. 1c). Injuries introduced 2 dpf on both sides of an injury inflicted before 30 hpf were completely restored despite disruption of duct patency and fluid flow by the initial lesion (Fig. 1c). Despite the remarkable difference in repair capacity between 1- and 2-day-old zebrafish embryos, the pronephros did not reveal obvious morphological differences between these two developmental stages, consistent with the observation that pronephric tubule formation is largely completed by 24 hpf[13]. Pronephric tubules at both stages displayed a defined apical-basolateral polarity, formed asymmetrical cellular appendages including cilia, and were surrounded by extracellular matrix that stained positive for laminin (Supplementary Fig. 1). However, one difference between the two repair responses was the accumulation of actomyosin bundles observed in 1-, but not in 2-day-old zebrafish embryos immediately adjacent to the injured segment (Fig. 1d, e). These findings define a novel developmental switch between two different repair mechanisms: injured pronephric ducts before 30–36 hpf are repaired by the contraction of actomyosin bundles and a purse-string-like occlusion (Fig. 1f); injuries at later time points trigger a migratory response that transiently overrides the posterior-to-anterior directed cell migration, re-establishing the integrity of the pronephros.

Utilizing a zebrafish mutant with a mutation in *ift88* (*oval*), required for normal ciliogenesis[14,15], we excluded a role of cilia in pronephros regeneration. The mutant zebrafish embryos exhibited defective cilia and pronephric cyst formation, but retained their regenerative capacity and re-established the integrity of the pronephros after injury (Supplementary Fig. 2). In the region of regeneration, no up-regulation of *n-myc*, *vim1*, *snail1*, *netrin*, *mmp9*, or markers for de-differentiation, such as *pax2.1* and *tbx2b*, was observed (Supplementary Fig. 3), supporting the absence of epithelial-to-mesenchymal transition during the repair process. To assess the involvement of the Wnt signaling pathway, we over-expressed the Wnt inhibitor *dkk1* in zebrafish embryos. Despite extensive expression of *dkk1*, regeneration of the injured pronephros was not impaired (Supplementary Fig. 4a). Furthermore, we did not detect an up-regulation of Wnt/ß-catenin-dependent gene transcription in the regenerating tissue, using a reporter transgenic fish line with a TCF-responsive element driving RFP expression[16] (Supplementary Fig. 4b). In addition, neither knockdown of zebrafish *celsr1*[17], a core component of the non-canonical Wnt/planar cell polarity pathway, nor depletion of *par6b*, a component of the apico-basal polarity complex, interfered with the repair process despite extensive morphological abnormalities of the pronephros, including cyst formation and duct dilatations (Supplementary Fig. 4c)[18]. All injuries ≤50 µm, and 80% of injuries ≤100 µm, corresponding to 10–12 cell diameters, were repaired (Supplementary Fig. 4d). Thus, the

migratory repair response represents a robust program that is not readily disturbed by interference with developmental signaling cascades.

**The absence of *cxcl12a* or *cxcr4b* results in defective repair**. The injury-induced migratory repair response briefly disrupted the posterior-to-anterior cell migration, and induced a bidirectional migration of the neighboring cells to close the injured gap (Fig. 2). Wounding reversed the direction of the cells at the anterior border of the gap, and forced them to migrate backwards, while the cells on the posterior site of the wound continued their migratory trajectory to close the gap (Fig. 2a, and Supplementary Movies 1 and 2). Ablating cells upstream and downstream of a cluster/grouping of cells further highlighted the dominant effect of the injury on the posterior-to-anterior collective cell migration present during this stage of development. At the border, surviving cells rapidly stretched to cover the wound regardless of its localization, whereas cells centered in the cluster ceased to migrate until the wound on both sides was closed (Fig. 2b). Distant from the repair, cells continued to migrate collectively in a posterior-to-anterior direction, whereas cells involved in repair moved more rapidly, often exceeding twice the speed if reversing direction (Fig. 2c). To identify molecules involved in the repair process, we compared the expression profile of micro-dissected 1- and 2-day-old zebrafish pronephric tubules (Supplementary Fig. 5). The Zebrafish Information Network (ZFIN) database was examined for candidate genes previously detected in the pronephros by in situ hybridization, yielding three genes, *cxcl12a*, *myca*, and *mafba*. Since *mafba* is primarily expressed in neurons, rhombomeres and the pancreas[19], we analyzed the role of zebrafish Cxcr4b and Cxcl12a during the collective cell migration of pronephric cells, using a *cxcr4b:H2B-RFP*; *cldnb:lynGFP* double transgenic zebrafish with RFP-labeled nuclei and GFP-labeled membranes of pronephric duct (Fig. 3a, and Supplementary Movies 3–5). The nuclei of migrating pronephric cells were tracked at 36 and 41 hpf for 3 h, and at 48 hpf for 8 h. The average track speed and displacement length of approximately 40 nuclei was calculated for control, *cxcl12a* and *cxcr4b* mutant zebrafish embryos (Fig. 3b, c). The average track speed for all genotypes and time points was $7.12 \pm 0.19$ μm/h, consistent with the previously reported migration speed[3]. There were no detectable differences between control and *cxcl12a*- or *cxcr4b*-deficient zebrafish embryos, revealing that in contrast to their essential role in pLL development, neither gene product is required for the collective cell migration during zebrafish pronephros development. However, in *cxcr4b*-deficient zebrafish embryos approximately 60% of injuries (Fig. 4a), and in cxcl12-deficient zebrafish embryos approximately 40% of injuries were not repaired (Fig. 4b). The repair responses were characterized by complex abnormalities. The patency of the pronephric duct was often not re-established due to a lack of coordinated migratory responses. Curiously, the migrating neighboring tubular epithelial cells deviated from the normal anterior–posterior axis, and formed dorsal pouches (Fig. 4c–j). Cxcr4b was detectable only at low levels during normal collective cell migration of zebrafish tubular epithelial cells, (Supplementary Fig. 6a–f). However, time-lapse video-microscopy of a fluorescently tagged Cxcr4b expressed under its own promoter (Fig. 5a, and Supplementary Movies 6,7) and in situ hybridization (Fig. 5b, c) revealed an up-regulation of *cxcr4b* expression in tubule cells adjacent to the injury. While *cxcl12a* was expressed in the Zebrafish pronephros 2 dpf (Supplementary Fig. 6g)[20], heat shock and ectopic production of Cxcl12a in *hsp70:cxcl12a* zebrafish embryos constrained the repair response (Supplementary Fig. 6h), further supporting a

role of the Cxcl12a/Cxcr4b signaling pathway in the immediate response to injury.

**Zebrafish *myca* is required for a normal repair response**. Zebrafish *myca* was prominently expressed along the pronephros and the cloaca at 24–72 hpf (Supplementary Fig. 7a–c), and up-regulated in the tubular epithelial cells adjacent to a laser-induced wound (Supplementary Fig. 7d). Knockdown of *myca* with a splice-blocking morpholino oligonucleotide (MO) impaired the repair in 2-day-old zebrafish embryos, revealing a role for *myca* in the repair process; similar results were obtained for a translation-blocking MO (TBM) (Supplementary Fig. 8). Co-expression of *myca* mRNA partially rescued the repair defect, while inactive mRNA had no significant effect. Time-lapse imaging demonstrated that the pronephric duct cells next to the injury remained stationary, or failed to establish the patency of the duct despite making contact with the corresponding end of the tubule (Supplementary Fig. 9a,b, and Supplementary Movies 8,9), suggesting that *myca* is required to transiently override the posterior-to-anterior cell migration. Since the interpretation of MO-induced phenotypic changes is notoriously difficult[21], we generated a *myca* mutant zebrafish line, lacking 20 base pairs in the 5′ region of exon 2 (Supplementary Fig. 10). While all heterozygote and wild-type *myca* zebrafish embryos repaired, 25% of the *myca*$^{-/-}$ zebrafish displayed a repair defect ($p = 0.005$), confirming the *myca* MO results. To further validate these results and to obtain insight into the molecular mechanisms controlling the repair process, we confirmed the zebrafish results in mice with an inducible, kidney-specific deletion of *Cxcl12* and *Myc*.

***Cxcl12* or *Myc* deficiencies delay recovery after injury**. CXCL12 and CXCR4 are expressed in the kidney and up-regulated in response to injury[22–28]. To delineate the intrinsic, kidney-specific role of the CXCL12/CXCR4 signaling during the early recovery of mammalian kidneys from ischemia/reperfusion (I/R) injury, we used the doxycycline-inducible *Pax8rtTA*TetOCre* transgene to drive kidney-specific excision of floxed *Cxcl12* alleles. Male mice at the age between 6 and 8 weeks with the genotype *Cxcl12*$^{fl/fl}$*Pax8rtTA*TetOCre*, and male control mice were treated for 2 weeks with doxycycline, followed by a wash-out period of 1 week to avoid doxycycline-mediated artifacts[29]. There were no functional or histological differences between control and knockout mice at the end of the washout period (Supplementary Fig. 11). Mice were then subjected to bilateral I/R injury. To detect differences in early recovery, mice were analyzed 12 h after I/R injury. Mice lacking kidney-specific *Cxcl12* displayed higher serum urea levels and higher urinary NGAL excretion than control mice (Fig. 6a–c). While histological analysis showed the typical characteristics of severe damage, obvious morphological differences were not apparent between control and knockout mice 12 h after injury (Fig. 6d, and Supplementary Fig. 12), indicating that the different outcomes resulted from early adaptive changes rather than structural differences detectable by histology.

To examine the role of *Myc* in early recovery after I/R injury, we used the doxycycline-inducible *Pax8rtTA*TetOCre* transgene to drive kidney-specific excision of floxed *Myc*. Male mice at the age between 6 and 8 weeks with the genotype *Myc*$^{fl/fl}$*Pax8rtTA*TetOCre* and male control mice were treated for 2 weeks with doxycycline, followed by a wash-out period of 1 week. There were no functional or histological differences between control and knockout mice at the end of the washout period (Supplementary Fig. 13). Mice were then subjected to I/R injury. To detect differences in early recovery, mice were analyzed 12 h after I/R injury (Fig. 6e). Mice lacking *Myc* displayed higher serum urea

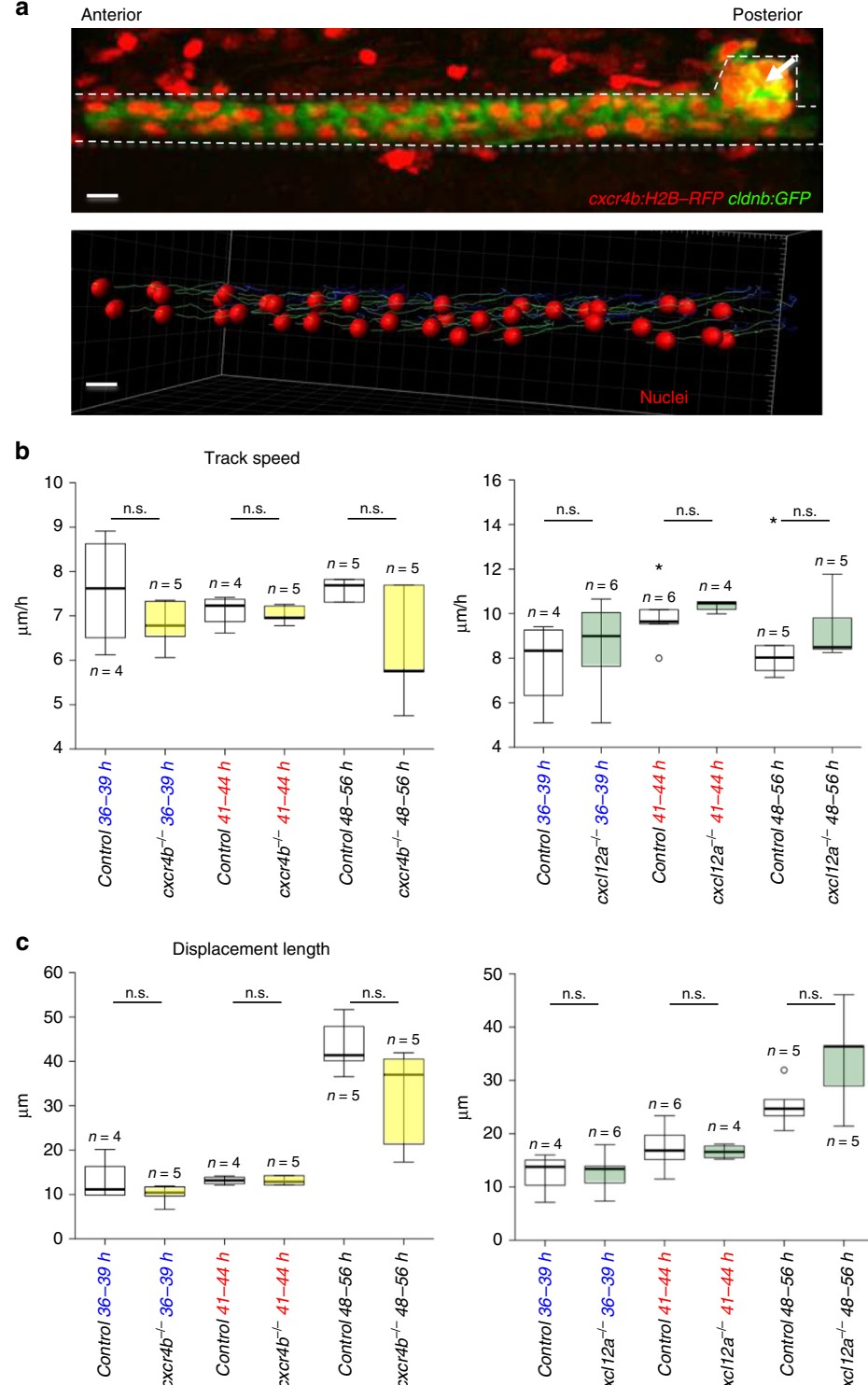

**Fig. 3** *cxcr4b* or *cxcl12a* deletion does not affect pronephros cell migration. **a** Collective cell migration in the zebrafish pronephros was quantified, using a *cxcr4b:H2B-RFP; cldnb:GFP* double transgenic zebrafish line (upper panel, see Supplementary Movie 4). The transgene *cxcr4b:H2B-RFP* labels the nuclei of cells that express RFP under the control of the endogenous *cxcr4b* promoter; *cldnb:GFP* labels the membranes of pronephric duct cells. The pronephros and the corpuscle of Stannius are outlined with dashed lines. The arrow points to the corpuscle of Stannius (scale bars, 10 μm). A model of collective cell migration was generated (lower panel), using the image tool Imaris. The migration of ca. 40 pronephric nuclei was tracked over 8 h, starting 48 hpf. The tracks for each cell were calculated and represented as lines. The observation times were coded by blue-to-green colors; blue are early time points and green are later ones (see Supplementary Movie 5). **b** The mean track speed and displacement length, i.e., the length that a cell traveled over the observation period were measured at 36 and 41 hpf for 3 h, and at 48 hpf for 8 h. Deficiency of *cxcr4b* (left panel) or *cxcl12a* (right panel) did not affect track speed (left panel). **c** Deficiency of *cxcr4b* (left panel) or *cxcl12a* (right panel) did not affect displacement length. Note that the displacement length is larger in the 48–56-hour-interval, because cells were tracked for a longer time period (8 h) in comparison with the earlier time points (3 h each). In contrast, the speed did not change significantly over time (n.s., not significant; *t*-test)

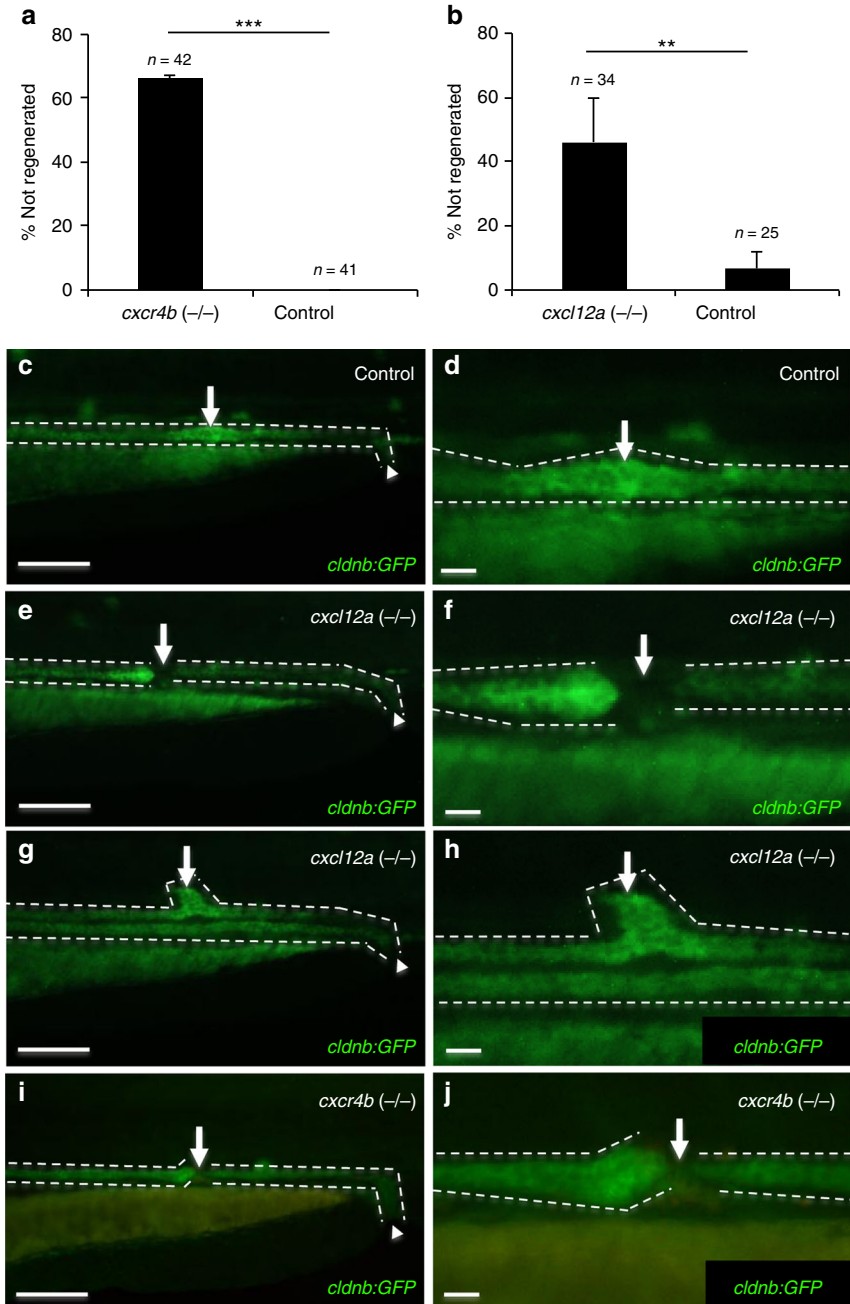

**Fig. 4** Defective repair in *cxcl12a* (−/−) and *cxcr4b* (−/−) zebrafish embryos. **a** While control zebrafish embryos repair a lased-induced injury, more than 60% of *cxcr4b*-deficient zebrafish embryos (means ± SEM; ***$p < 0.001$; *t*-test) and **b** more than 40% of *cxcl12a*-deficient zebrafish embryos did not repair the wound, but instead revealed various abnormalities (means ± SEM; **$p < 0.01$; *t*-test). **c** The pronephros of 2-day old *wild-type cldnb:GFP* control zebrafish embryos rapidly regenerates after a laser-induced injury. The arrow points to the regenerated area at 24 h after the injury. The arrowhead points to the cloaca. The pronephros is outlined with dashed lines (scale bar, 100 μm). **d** Magnification of the regenerating region depicted in **c** (scale bar, 20 μm). **e** *cxcl12a* (−/−) zebrafish embryos fail to regenerate and the pronephric tubule remains discontinued. The arrow points to the gap in the tubule. The pronephros is outlined with discontinued lines. The arrowhead points to the cloaca. Note the dilation of the anterior duct due to ongoing filtration (scale bar, 100 μm). **f** Magnification of the region that failed to regenerate depicted in **d** (scale bar, 20 μm). **g** In some cases, *cxcl12a* mutant embryos showed aberrant regeneration and formed dorsal pouches. The arrow points to the abnormally regenerated tubule. The pronephros is outlined with discontinued lines. The arrowhead points to the cloaca (scale bar, 100 μm). **h** Magnification of the abnormally regenerated region depicted in **e** (scale bar, 20 μm). **i** *cxcr4b* mutant zebrafish embryos failed to re-establish the tubular patency similar to *cxcl12a* mutant embryos. The arrow points to the gap in the tubule. The pronephros is outlined with discontinued lines. The arrowhead points to the cloaca (scale bar, 100 μm). **j** Magnification of the region that failed to regenerate depicted in **f** (scale bar, 20 μm)

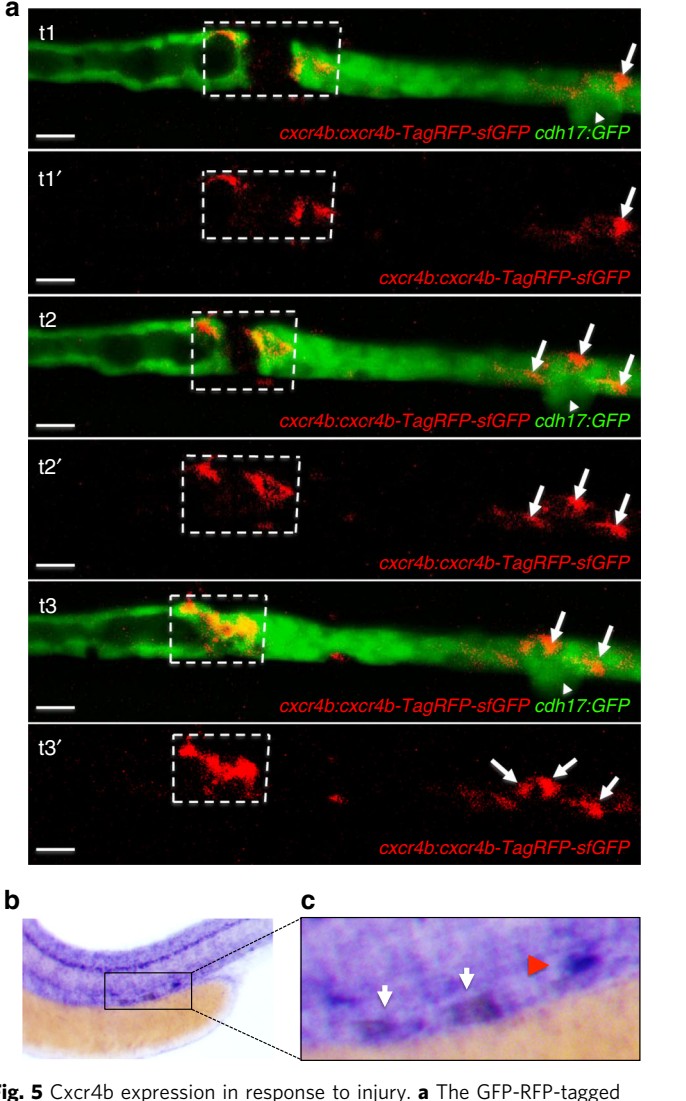

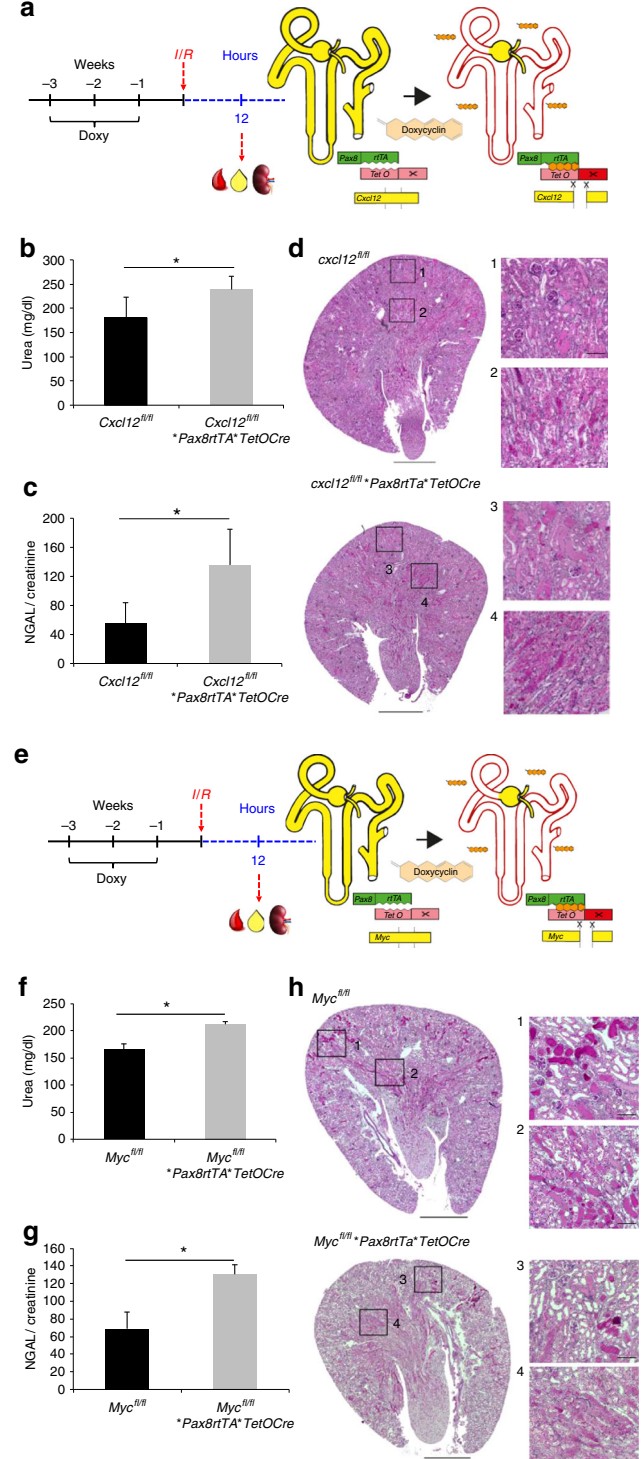

**Fig. 5** Cxcr4b expression in response to injury. **a** The GFP-RFP-tagged Cxcr4b (*cxcr4b:cxcr4b-tFT*) was up-regulated in cells adjacent to the laser-induced injury. The arrowhead points to the corpuscle of Stannius. Arrows point to RFP-positive cells, accumulating distally after the injury. Sequential frames were taken from a time-lapse video at t1 = 96 min, t2 = 120 min, and t3 = 160 min after injury (scale bar, 20 μm). **b** In situ hybridization revealed up-regulated *cxcr4b* expression in the leading ends of the regenerating tubule 2 h after laser-induced ablation. The *cxcr4b* up-regulation is outlined in a dashed box. Note that *cxcr4b* is strongly expressed in the corpuscle of Stannius (arrow) and in the lateral line. **c** Magnification of the injury region depicted in **b**. The white arrows point towards the up-regulated *cxcr4b*, the red arrowhead towards the corpuscle of Stannius

levels and higher urinary NGAL excretion than control mice (Fig. 6f, g), while histological differences were not apparent between these two mouse strains (Fig. 6h, and Supplementary Fig. 14).

Although only a small compartment is affected by the tissue-specific gene excision, RNA sequencing of kidneys removed from control and KO mice 12 h after I/R injury showed a clear separation of KO and WT mice in the principal component analysis (Supplementary Fig. 15). Both KO kidneys displayed reduced metabolic and mitochondrial-associated gene expression profiles (Supplementary Fig. 16–18, and Supplementary Data 1–4); this metabolic signature was reflected in the top 25 down-

regulated Gene Ontology (GO) terms, using Gene Set Enrichment analysis (GSEA) (Supplementary Fig. 19). GO term enrichment analysis also revealed significant changes in retinoic acid metabolic processes for the *Cxcl12* KO ($q = 6.8 \times 10^{-6}$) and for the *Myc* KO ($q = 0.006$) kidneys exposed to I/R injury (Fig. 7a). Retinoic acid can induce *Cxcr4*, enhance mitochondrial function[30], and augment energy metabolism[31], providing a potential approach to reverse the observed defects. We therefore tested whether retinoic acid can ameliorate the recovery after I/R injury.

**Fig. 6** *Cxcl12* or *Myc* deletion impairs the recovery after injury. **a** Male mice with the genotype *Cxcl12^fl/fl*Pax8rtTA*TetOCre* (*n* = 5) and control mice (*n* = 5) with the genotype *Cxcl12^fl/fl*Pax8rtTA* were subjected to I/R injury. Histology and biochemical analysis were performed 12 h later. **b** Serum urea concentrations were significantly higher in mice lacking *Cxcl12* (*$p < 0.05$). **c** Urinary NGAL excretion, normalized for urinary creatinine, was significantly higher in mice lacking *Cxcl12* than in the control animals (means ± SEM; *$p < 0.05$; *t*-test). **d** PAS-stained kidney sections from *Cxcl12^fl/fl*Pax8rtTA* (controls) and *Cxcl12^fl/fl*PAX8rtTA*TetOCre* mice were obtained 12 h after I/R injury. Squares labeled 1 and 3 show enlarged images of the cortex, squares labeled 2 and 4 show enlarged images of the cortico-medullary junction. **e** Male mice with the genotype *Myc^fl/fl*Pax8rtTA*TetOCre* (*n* = 5) and control mice (*n* = 5) were subjected to I/R injury. Histology and biochemical analysis were performed 12 h later. **f** Serum urea concentrations were significantly higher in mice lacking *Myc* (*$p < 0.05$). **g** Urinary NGAL excretion, normalized for urinary creatinine, was significantly higher in mice lacking *Myc* than in the control animals (means ± SEM; *$p < 0.05$; *t*-test). **h** PAS-stained kidney sections from *Myc^fl/fl*Pax8rtTA* (controls) and *Myc^fl/fl*PAX8rtTA*TetOCre* mice were obtained 12 h after I/R injury. Squares labeled 1 and 3 show enlarged images of the cortex, squares labeled 2 and 4 show enlarged images of the cortico-medullary junction (scale bars, 1 mm and 100 μm)

While tretinoin in doses between 0.1 and 10 μM increased CXCR4 levels in Jurkat cells (Supplementary Fig. 20), both control and *Myc* KO mice benefited from tretinoin (40 mg/kg) applied 12 h before and during I/R injury (Fig. 7b, c).

***Cxcl12* and *Myc* promote glycolysis.** To obtain additional insight into the mechanisms responsible for the detrimental consequences of *Cxcl12* and *Myc* deletion, we analyzed the urine samples of control and knockout mice at 12 h after I/R injury. The principal component analysis grouped control and knockout animals into distinct clusters on the basis of their urine metabolites (Supplementary Fig. 21); key metabolites responsible for this separation were α-hydroxy-isovalerate, leucine, isoleucine, lactate, citrate, and glucose. While branched chain amino acids (e.g., leucine, isoleucine, and α-hydroxy-isovalerate) were increased in the urine of *Cxcl12*- and *Myc*-deficient kidneys, lactate and glucose were increased in the urine of control animals, indicating compromised glycolysis and gluconeogenesis in *Cxcl12*- and *Myc*-deficient mice subjected to I/R injury (Fig. 8). To substantiate our hypothesis, we isolated tubular epithelial cells from wild-type and *Myc^fl/fl*Pax8rtTA*TetOCre* mice. Both cell types were exposed to doxycycline (0.5 μg/ml) for 24 h, followed by a 24-h incubation period without doxycycline to avoid doxycycline-dependent metabolic effects; depletion of *Myc* in cells containing the floxed *Myc* allele was confirmed by Western blot analysis (Supplementary Fig. 22a). To compare the glycolytic capacity between these two primary cell types, the extracellular acidification rate (ECAR) was measured after sequential addition of glucose, oligomycin and 2-deoxy-D-glucose (2-DG). Non-glycolytic acidification was slightly higher in tubular epithelial cells lacking *Myc*. However, both basal glycolysis, measured after addition of glucose, and maximal glycolytic capacity, determined after addition of oligomycin, were impaired in tubular epithelial cells, lacking *Myc* (Supplementary Fig. 22b). Addition of 2-DG confirmed that glycolysis was responsible for the changes in ECAR. Thus, Myc deficiency compromises the ability of tubular epithelial cells to mount a maximal glycolytic response after ischemia.

We speculated that glycolysis may contribute to fast cell movements during development and repair mechanisms requiring rapid adaptive responses, and tested this hypothesis in the zebrafish pLLP and pronephros model. The small molecule 3-(3-pyri-dinyl)-1-(4-pyridinyl)-2-propen-1-one (3PO) is a phospho-fructokinase-2/fructose-2,6-bisphosphatase inhibitor that partially and transiently reduces glycolysis in vivo[32]. Applied at a low concentration of 20 μM, 3PO does not cause developmental defects[32], but slowed the migration of the pLLP and pronephros repair after injury. In contrast, 3PO had no effect on the collective cell migration of the pronephros (Fig. 9). While 2-DG applied at a concentration of >100 μM was well tolerated and had no detectable effect on zebrafish embryogenesis, 2-DG applied at concentrations of 40 and 80 μM slowed pLLP migration and pronephros repair, but did not affect collective cell migration of the pronephros (Fig. 9). 3PO and 2-DG delayed but did not prevent pronephros repair, suggesting that both compounds do not disrupt but rather slow the migratory response after injury. Together, these findings suggest that anaerobic energy production, enabled by CXCL12 and MYC, facilitates fast cell movements during development and repair processes.

**Discussion**

The zebrafish pronephros model revealed the existence of two different injury responses: a purse-string pronephros occlusion mediated by actomyosin bundles during early embryogenesis and a migration-based pronephros repair at later developmental stages. Cell plasticity during early development usually ensures full recovery from non-fatal injuries. Thus, it was unexpected that pronephric tubules lack the migration-based repair mechanism during early embryogenesis although morphologically normal appearing pronephric tubules are established within the first 24 h. Instead ablation of tubular epithelial cells before 30 hpf triggers actomyosin bundles in neighboring cells that irreversibly occlude the lumen of pronephric tubules. This occlusive response may prevent electrolyte and water disturbances during critical stages of development, but persists during later embryogenesis, and results in irreversible loss of the injured tubule. Fluid transport and intraluminal flow due to active solute transport across the tubular epithelial cells is detectable as early as 24 hpf[3]. Since collective cell migration accelerates with the onset of fluid flow, it was proposed that fluid flow supports the collective migration of tubular epithelial cells[3]. However, pronephros injuries sequentially applied before 30 hpf and after 36 hpf revealed that injuries introduced downstream of an occlusive injury were repaired by a migratory repair with similar dynamics as injuries upstream of the occlusion. Furthermore, deletion of *Ift88*, which reduces intra-luminal fluid flow due to ciliary defects and induces proximal cyst formation[15], did not affect the repair response, indicating that the injury-induced migration occurs largely independent of fluid flow. Thus, the time between 30 and 36 h after fertilization defines a developmental switch between two different repair programs in response to tubular cell necrosis.

*Cxcl12a* and *Cxcr4b* are essential for normal collective cell migration and development of the pLL[33]. Loss of *Cxcr4b* results in uncoordinated cell movements and greatly reduced net cell displacement, while loss of *Cxcl12a* leads to non-directed, random cell migration[34]. In contrast to their essential role during the lateral line development, neither molecule is required for collective cell migration during zebrafish pronephros development. Measurement of cell migration and cell displacement revealed no differences between control and *cxcr4b*- or *cxcl12a*-deficient zebrafish embryos. Zebrafish Cxcr4b was expressed at low levels in the developing pronephros, and only detectable, using a stabilized membrane-version of Cxcr4b that fails to undergo ligand-dependent internalization and degradation[7]. However, after laser-induced injury, *cxcr4b* was up-regulated in pronephric duct cells adjacent to the injury, and expression appeared to increase during

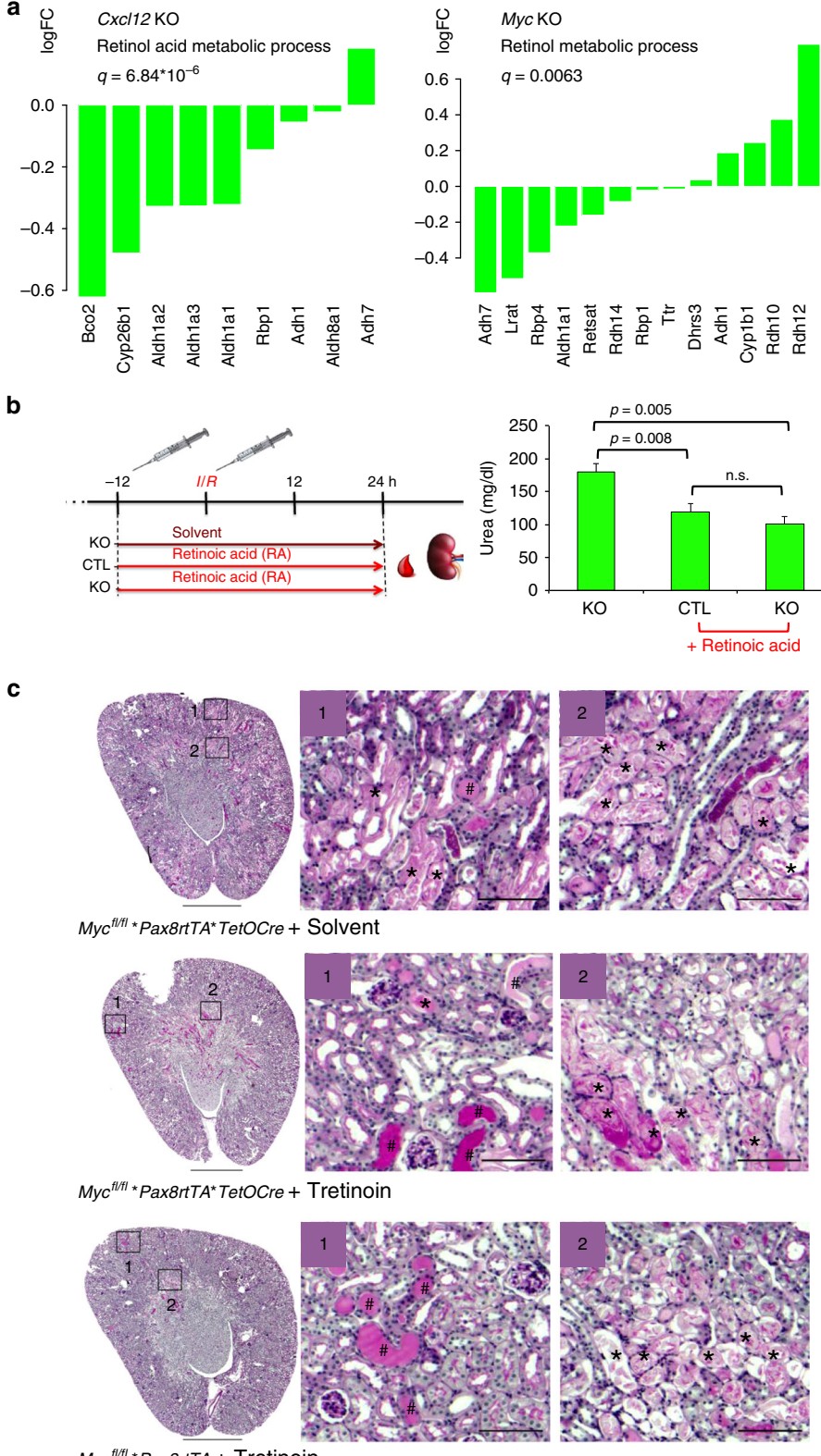

**Fig. 7** Retinoic acid accelerates the recovery after injury. **a** GO term analysis of *Cxcl12*[fl/fl]- and *Myc*[fl/fl]*Pax8rTA*TetOCre* compared to control mice revealed suppressed retinol acid metabolic and retinol metabolic processes, respectively; the individual genes of the GO term are depicted. **b** *Myc*[fl/fl]*PAX8rTA/TetOCre* (KO) (*n* = 13) and *Myc*[fl/fl]*PAX8rTA* (control animals) (*n* = 7), lacking the Cre recombinase were treated with doxycycline for 2 weeks, followed by a washout period of 1 week. Tretinoin (RA, 40 mg/kg) was applied 12 h before and during the ischemia/reperfusion injury to seven control and six KO mice. Eight KO mice were injected with an equal volume of cottonseed oil, only (solvent). One of the untreated animals was removed from further analysis due to bleeding and incomplete clamping. Urea levels were measured 24 h after surgery. Differences between mice treated with RA and untreated animals were statistically significant (means ± SEM; *p < 0.05; *t*-test), while there were no differences between RA-treated control and *Myc* KO mice. **c** Histology of the cortical and medullary region revealed significant tubular injury with intraluminal debris (*) and casts (#) in all three groups after ischemia/reperfusion injury (scale bars, 1 mm and 100 μm)

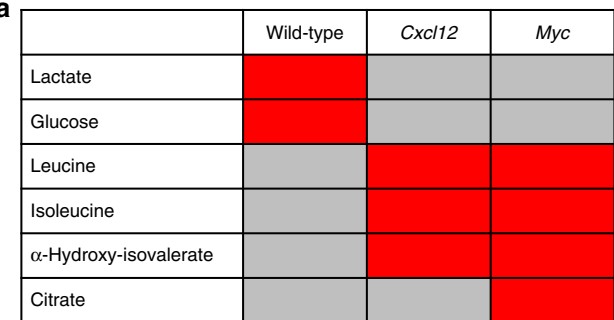

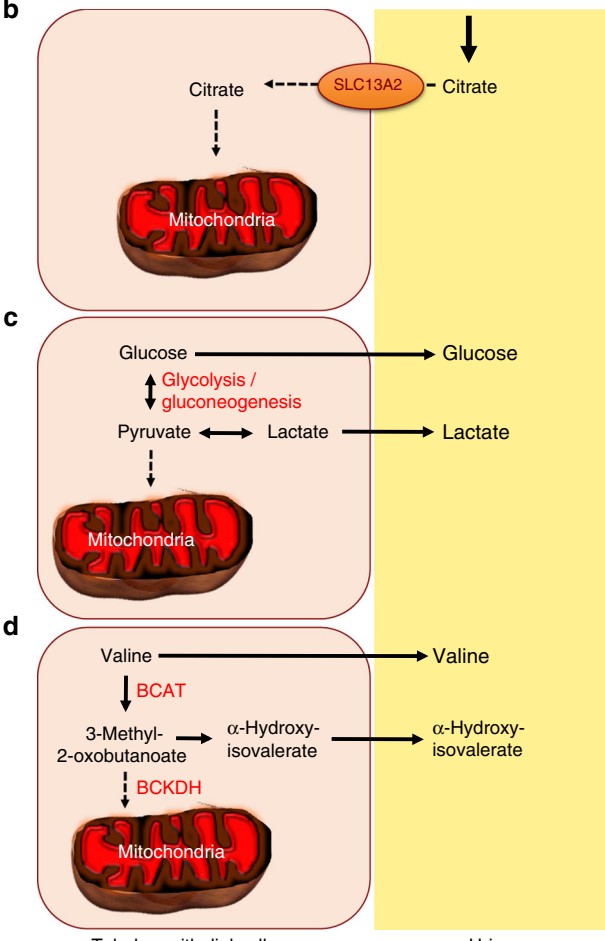

**Fig. 8** Key metabolite changes after *Cxcl12* or *Myc* deletion. **a** While lactate and glucose were elevated after ischemia/reperfusion (I/R) injury in the urine of wild-type animals (red box), leucine/isoleucine, α-hydroxy-isovalerate and citrate were elevated in the urine of knockout (KO) animals. **b** In humans, approximately 20 mmol of citrate are filtered daily; 70–90% of the filtered citrate are reabsorbed by di- and tricarboxylate transporters (e.g., NaDC1/SLC13A2), and metabolized by mitochondria. Slowing entry of citrate into mitochondria due to mitochondrial dysfunction reduces tubular citrate uptake, contributing to increased urinary citrate concentrations[67, 68]. **c** Lactate production is rapidly increased during renal ischemia due to enhanced glycolysis[46]. The efflux of lactate drives glycolysis and maintains intracellular pH, increasing urinary lactate concentrations. Defective glycolysis results in reduced lactate production and lactate efflux in KO animals. Gluconeogenesis is enhanced during early reperfusion, resulting in increased glucose production and urinary glucose concentrations in wild-type mice, while defective pyruvate and lactate production during ischemia prevents gluconeogenesis in KO animals. **d** Knockout of *Cxcl12* or *Myc* compromises mitochondrial catabolism of branched-chain amino acids (e.g., valine), resulting in increased accumulation of α-hydroxy-isovalerate. The transamination by branched-chain amino acid transaminase (BCAT) as well as the conversion to α-hydroxy-isovalerate by lactate dehydrogenase occur in the cytoplasm, while the rate-limiting step of aerobic catabolism of branched-chain amino acids (valine, leucine, isoleucine) by branched-chain alpha-keto acid dehydrogenase (BCKDH) occurs in mitochondria[69]. Thus, defective mitochondrial functions cause increased leucine, isoleucine, and α-hydroxy-isovalerate concentrations in the urine[70]

required for the slow mode of collective cell migration observed during pronephros development, but becomes essential for rapid adaptive processes after injury. Deletion of *Cxcl12* or *Cxcr4* in mice, while causing defective angiogenesis and glomerular capillary development, also does not affect the development of the tubular segments[35], reflecting the dispensability of Cxcl12a and Cxcr4b during normal zebrafish pronephros development. Based on its well-established function as a chemokine that controls homing of circulating hematopoietic cells, it has been postulated that CXCL12 is released from damaged kidneys to attract hematopoietic stem cells to the site of renal damage[36]. CXCL12 and CXCR4 are rapidly induced in renal cells next to necrotic regions[23,28]. However, since genetic labeling and tracking studies have not revealed an integration of extra-tubular cells into damaged kidney tubules[37], these observations argue against a role of CXCL12 as long-range chemo-attractant. To assess the contribution of CXCL12 expressed by tubular epithelial cells, we utilized a doxycycline-inducible Pax8-mediated excision of floxed *Cxcl12*. Elimination of *Cxcl12* delayed the recovery after IR injury. Since recruitment of extra-tubular cells seems an unlikely event[38], this observation suggests that CXCL12 is needed for early repair responses.

Gene expression profiling of micro-dissected zebrafish pronephric tubules also identified *myca* as molecule involved in the repair response. MYC belongs to a family of transcription factors that regulates the expression of a large number of target genes, controlling almost every aspect of cell proliferation, differentiation, and metabolism. MYC promotes cell migration and invasion, for example by activating Rho GTPases[39], but can also display anti-migratory effects through repression of CCL5/RANTES[40]. Since MYC enhances the promoter activity of CXCR4[41], we examined whether Cxcr4b can act down-stream of *myca* in the zebrafish pronephros repair model. Knockdown of *myca*, the zebrafish homologue of *Myc*, caused a repair defect that was partially rescues by *cxcr4b* mRNA, indicating that Myca acts in part through inducing *cxcr4b* expression to promote the

approximation of the two pronephric duct ends surrounding the injured gap. Consistent with a requirement for Cxcl12a/Cxcr4b signaling, various repair defects were observed in *cxcl12a*- or *cxcr4b*-mutant embryos, including a dorsal deviation of the repairing pronephric ends from the normal anterior–posterior axis and stationary, non-migratory tubular epithelial cells incapable of re-establishing the patency of the injured ducts. Ectopic Cxcl12a prevented the migration-dependent repair response, further supporting the critical role of Cxcl12/Cxcr4b signaling in response to pronephros injuries.

One difference between collective cell migration of the pLLP and tubular epithelial cells is the migration speed, which is almost 10 times higher for pLLP migration (~66 μm/h)[34] than for the collective cell migration of pronephric duct cells (2–8 μm/h). This difference suggests that Cxcl12a/Cxcr4b signaling may not be

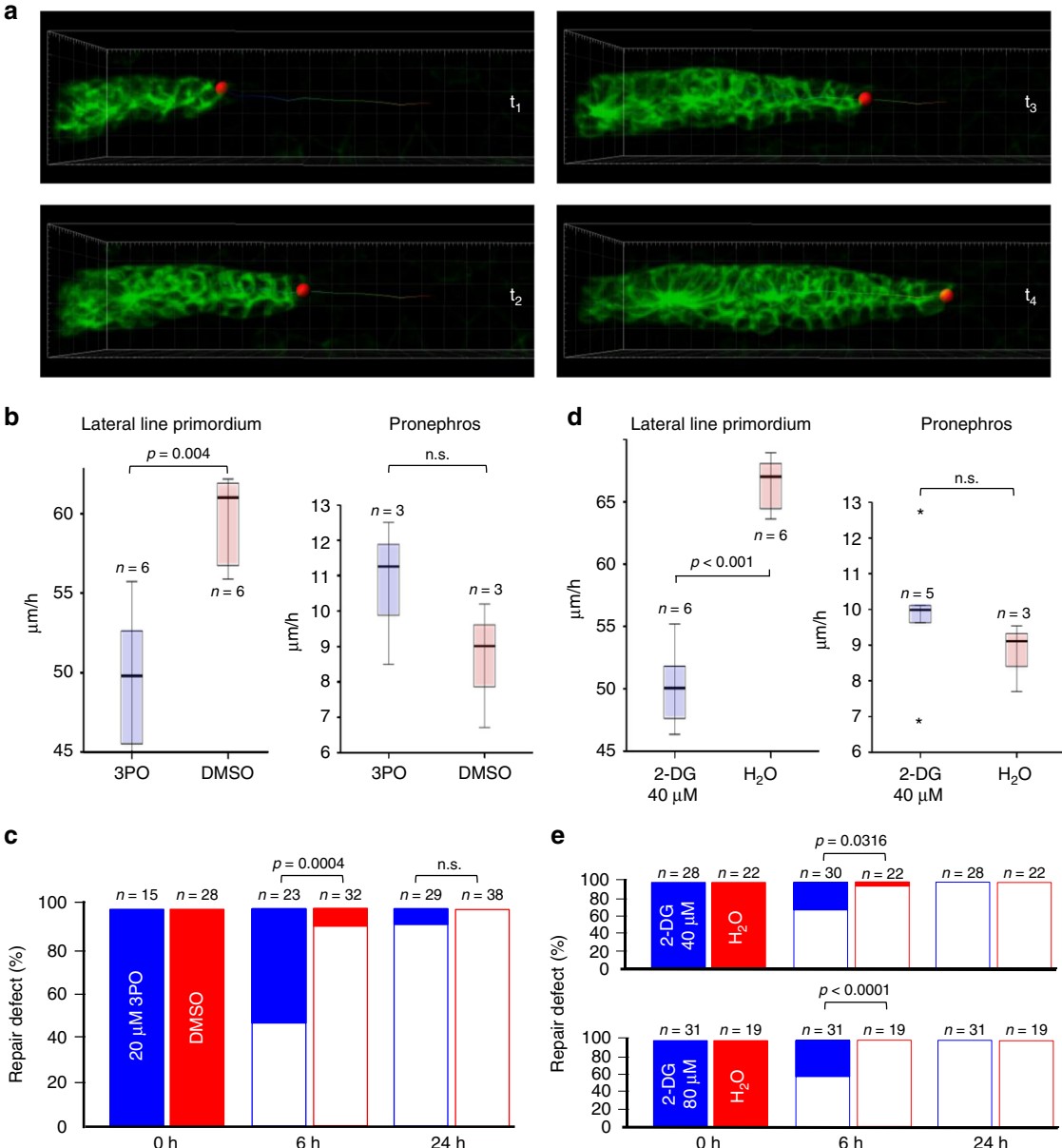

**Fig. 9** Inhibition of glycolysis slows migration. **a** Depicted are frames from Supplementary Movie 10, demonstrating the analysis of migration speed in the posterior Lateral Line Primordium (pLLP). A similar imaging approach was used to determine the speed of migrating cells in the zebrafish pronephros. **b** The small molecule 3-(3-pyri-dinyl)-1-(4-pyridinyl)-2-propen-1-one (3PO) at concentrations that had no recognizable developmental effects (20 μM) delayed pLLP migration, but did not affect pronephros cell migration. The box plots represent the median, the first and third quartile (boxed area), and 1.5× interquartile range (whiskers) (n.s., not significant; $t$-test). **c** 3PO delayed the repair of a pronephros injury determined at 6 h, but no effect was observed 24 h after the injury (Fisher's exact test). **d** The glucose analog 2-deoxy-D-glucose (2-DG) at 40 μM slowed pLLP migration similar to 3PO, but had no effect on pronephros collective cell migration. The box plots represent the median, the first and third quartile (boxed area), and 1.5× interquartile range (whiskers) (n.s., not significant; $t$-test). **e** While almost all injuries were repaired 6 h after wounding of control embryos, 40 μM 2-DG reduced the number of repaired injuries to 70% and 80 μM 2-DG to 58%. At 24 h after wounding, all injuries were repaired. Full bars depict not repaired injuries, open bars repaired injuries. The graphs represent the summary of at least three independent experiments (Fisher's exact test)

migratory response after a laser-induced pronephros injury. However, since the knockdown of *myca* did not prevent the injury-induced up-regulation of *cxcr4b*, up-regulation of *cxcr4b* does not require Myca. Thus, Myca may contribute to *cxcr4b* expression, but other transcription factors can likely compensate for the loss of *myca*.

Acute kidney injury (AKI) associated with renal failure is a frequent complication of severe human disease[42]. Despite advances in supportive care, AKI continues to be associated with

a high mortality. The repair process seems to occur instantaneously; however, the mechanisms initiating the repair process remain poorly understood since this process cannot be visualized in mammalian models of AKI. The repair response after a laser-induced injury of the zebrafish pronephric duct, captured by high-resolution microscopy and time-lapse imaging, provides the opportunity to identify molecules required for the regenerative responses after injury. This approach uncovered the role of *cxcl12a* and *myca* in early repair. Elimination of either molecule

in mice aggravated the recovery from I/R injury. Although the PAX8-driven, tubule-specific gene excision removed these molecules only from a limited subset of kidney cells, RNA sequencing of whole kidney lysates revealed extensive mitochondrial and metabolic changes in comparison to control mice exposed to the same injury, supporting recent observations that both CXCL12/CXCR4 and MYC signaling can control mitochondrial gene expression, and contribute to the adaptive responses after I/R injury[43,44]. Tubule-specific loss of *Cxcl12* and *Myc* also significantly reduced retinol metabolic processes, providing the rational to use retinoic acid to antagonize these signaling defects. Similar to *9-cis*-retinoic acid, a compound that ameliorates AKI through inhibition of NUR77[45], *all-trans*-retinoic acid (tretinoin) improved the renal function after I/R injury in both control and *Myc*-deficient mice, underlining the pleiotropic effects of this reagent on energy metabolism independent of *Myc*.

The zebrafish pronephros injury model suggests that Cxcl12a/Cxcr4b signaling is needed for fast adaptive responses after AKI. Loss of either *Cxcl12* or *Myc* in mice not only resulted in suppressed mitochondrial- and retinoic acid-associated gene expression, but also altered the metabolic composition of the urine, sampled 12 h after I/R injury. Hypoxia causes lactate accumulation in the kidney as the result of enhanced glycolysis and decreased oxidative phosphorylation; excess lactate is then rapidly excreted into the urine to maintain glycolysis and intracellular pH[46]. The decreased urinary lactate after *Myc* or *Cxcl12* deletion is therefore likely the result of diminished glycolysis. MYC stimulates glycolysis[47], and has recently been shown to control human pluripotent stem cell fate by maintaining a high glycolytic flux[48]. Since CXCL12/CXCR4 can facilitate glycolysis through stimulation of AKT[49], the absence of *Cxcl12* also reduces urinary lactate concentrations. The ability of MYC and CXCL12/CXCR4 signaling to maintain a high glycolytic flux may also allow tumor cells to invade tissues with low-oxygen tension[50,51].

Our findings confirm that glycolysis and anaerobic ATP generation represent an important energy source in response to I/R injury. However, the same metabolic switch is also utilized in developmental programs that require fast migratory responses. The ability of CXCL12 and MYC to promote cell migration by increasing glycolytic flux may also explain why CXCL12/CXCR4[52] and MYC[53] are associated with increased tumor invasiveness and poor survival.

## Methods

**Mouse models and ischemia reperfusion injury**. All animal experiments were approved by the local authorities (Regierungspräsidium Freiburg), and performed according to German law governing welfare of animals. Mice were housed with free access to chow and drinking water, and a 12-h day/night cycle. Standard procedures were used for breeding, maintenance, tail biopsy, and genotyping. *Cxcl12^fl/fl* C57BL/6J mice[54] obtained from The Jackson Laboratory (Bar Harbor, ME, USA), and *Myc^fl/fl* C57BL/6J mice were crossed with Pax8rTA and TetOCre animals[55]. All animals used in I/R injury experiments were 10–12 weeks old male mice and littermates. For *Cre* expression, animals and respective controls (lacking TetOCre) received doxycycline hydrochloride (Fagron) via drinking water (2 mg/ml with 5% sucrose (wt/vol), protected against light) for 14 days. After a 1 week-washout, renal I/R injury was performed under isoflurane anesthesia (4% isoflurane, 4 L/min flow for induction, followed by 2% isoflurane and 1 L/min flow for maintenance). Animals were kept on a heated surgical pad in supine position, and treated with the analgesic buprenorphine hydrochloride 0.05 mg/kg body weight subcutaneously before surgery. Renal pedicles were exposed via an abdominal midline incision. Renal arteries and veins were clamped for 22.5 min at 37 °C. After reperfusion, warm 0.9% NaCl was injected (40 μl/g body weight) intraperitoneally (i.p.), and metamizol was given for pain control via the drinking water (500 mg metamizol/100 ml water). Mice were sacrificed under ketamine/xyalzine anesthesia (120 and 8 μg/g body weight, i.p.), and urine and serum were collected 12 h after I/R injury. Both kidneys were perfused with PBS, and snap-frozen for RNA extraction, or immersion-fixed in 4% paraformaldehyde/PBS for further histological evaluation. For the tretinoin experiments, mice underwent right-sided nephrectomy followed by unilateral clamping of the left kidney[56]. Tretinoin (S1653, Selleckchem) was dissolved in cotton-seed oil (C7767, Sigma). Control animals were injected with an

equal volume of cotton-seed oil (solvent); one mouse of the control group was removed from further evaluation due to bleeding and incomplete clamping of the renal artery.

**Biochemical measurements**. Serum urea and urinary creatinine concentrations were measured using enzymatic colorimetric kits following the manufacturer's instructions (Lehmann). Optical densities for serum urea and urine creatinine were measured using a Genesys20 spectrophotometer (Thermo Fischer Scientific) and Tecan infinite M200 microplate reader, respectively.

**Mouse histology**. Kidneys were immersion-fixed in 4% phosphate-buffered paraformaldehyde and embedded in paraffin. Kidney sections of 4 μ thickness were used for Periodic Acid Schiff (PAS) staining. Images were taken using ZEISS Axio observer Z1 microscope, and analyzed using ZEN blue software.

**Western blot**. Urine volumes were normalized against creatinine concentration, and separated on 10% SDS-PAGE. Urinary NGAL was detected using a goat polyclonal antibody against mouse NGAL (AF1857, R&D Systems) at a 1:2000 dilution. CXCR4 in Jurkat cells was determined by Western blot analysis, using an anti-CXCR4 antibody at a 1:500 dilution (ab124824, Abcam). Tubulin at a 1:5000 dilution (T6557, Sigma-Aldrich) was used to control for loading. CXCR4 protein levels were quantified and normalized to γ-tubulin levels, using the LabImage software (Intas Science Imaging GmbH). MYC was detected by rabbit anti-MYC antibody (ab32072, Abcam) in primary renal cells, using the antibody at a 1:500 dilution. All uncropped Western blots can be found in Supplementary Fig. 23.

**RNA extraction from the kidney**. Snap frozen kidneys were homogenized in RLT buffer (Qiagen) supplemented with 1% 2-mercaptoethanol by using Ultra-Turrax (Ika T10 basic). Total RNA was extracted from homogenized samples according to manufacturer's instructions using RNeasy Mini Kit (Qiagen). The concentration and quality of the isolated RNA samples were measured by NanoDrop One (Thermo Scientific). Quality of RNA was further checked by Fragment analyzer and sent to GATC Biotech for RNA-sequencing.

**Statistical analysis**. Results were expressed as mean ± SEM. All data were tested for significance with unpaired Student $t$ test (mice and zebrafish), or Fisher's exact test (zebrafish), using GraphPad Prism 6 (GraphPad Software). The box plots (IBM SPSS Statistics 23) represent the median, the first and third quartile (boxed area), and 1.5× interquartile range (whiskers). Differences with values $p < 0.05$ were considered significant.

**Zebrafish embryo manipulation**. Zebrafish embryos were staged according to Kimmel et al.[57]. The following MOs (GeneTools) were used: *myca*-TBM (translation-blocking morpholino) 5′-AACTCGCACTCACCGGC ATTTTGAC-3′; *myca*-SBM (splice-blocking morpholino) 5′-CATTTTGACACTTGAGG AAG-GAGAT-3′; standard control MO (5′-CCTCTTACCTCAGTTACAATTTATA-3′); *p53* MO (5′-GCGCCATTGCTT TGCAAGAATTG-3′). Transgenic zebrafish lines *Tg(−8.0cldnb:lynEGFP)^zf106* (cldnb:GFP), cxcr4b:cxcr4b-tFT, cxcr4b:mem-tFT, cdh17:GFP, and mutant lines cxcr4b^t26035, cxcl12a^t30516, cxcr7b^sa16 have been published[7,58–60]. The transgenic line cxcr4b:H2B-RFP^fu13Tg was generated using BAC homologous recombination by replacing the second exon of cxcr4b in the cxcr4b-containing fosmid CH1073-406F3 by H2B-RFP followed by zebrafish transgenesis. Ectopic production of Cxcl12a was triggered by heat shock for 60 min before laser ablation, using the transgenic zebrafish line cdh17:GFP; hsp70:cxcl12a. Mutant myca mRNA (*myca mRNA) was generated by replacing four nucleotides (depicted in capital letters) adjacent to the translational start site 5′-atgccAg-tAagCgcAagtt-3′. To generate the mycaΔ20; wt1b:GFP; cdh17:GFP zebrafish line, 1 nl of a solution containing 150 ng/μl of myca gRNA (5′-GAGAGC-GACTGTCTCCGGCT-3′) targeting the second exon of myca and 343 ng/μl of Cas9 mRNA was injected into the yolk of wt1b:GFP; cdh17:GFP transgenic eggs. F0 carriers were outcrossed to wt1b:GFP; cdh17:GFP zebrafish, and the F1 progeny was screened for CRISPR-induced mutations by PCR with primers myca-ex2-f1 (5′-AAAATGCTGGTGAGTGCGAG-3′) and myca-ex2-r1 (5′-GCAGTCCTG-GATGATGATGG-3′). A heterozygous carrier with 20-bp deletion in the second exon of myca was identified and crossed with wt1b:GFP; cdh17:GFP zebrafish to establish the mycaΔ20; wt1b:GFP; cdh17:GFP line. The 20-bp deletion leaves the first 62 amino acids of myca intact, but results in a frame-shift and a premature stop codon after an additional 75 base pairs, encoding for 25 nonsense amino acids. The non-metabolizable glucose analog 2-deoxy-D-glucose (2-DG) (D8375, Sigma-Aldrich) was applied at concentrations of 40 and 80 μM starting 8 hpf. The small molecule compound 3-(3-pyridinyl)-1-(4-pyridinyl)-2-propen-1-one (3PO) (#525330, Calbiochem) was applied at a concentration of 20 μM starting 8 hpf.

**Zebrafish imaging**. Cell ablation was performed using 2-photon laser (Chameleon) mounted on LSM 510 Zeiss microscope (Carl Zeiss, Jena, Germany). Approximately 80 μm of the pronephros were ablated unless stated otherwise. Embryos were analyzed under a Leica MZ16 stereo-microscope (Leica, Solms, Germany). Differential interferences contrast (DIC) and non-confocal fluorescent

images were taken with a SPOT Insight Fire Wire System (Diagnostic Instruments, Sterling Heights, USA). For time-lapse movies the embryos were embedded in 1.5% low melting agarose. Confocal images were recorded on Zeiss LSM 510 with a C-Apochromat 40×/1.2 objective (Carl Zeiss, Jena, Germany). Confocal Z-stakes were acquired every 3 min, and time-lapse movies from the maximum Z-projections were assembled using LSM, ImageJ or Imaris software. Cell tracking was performed using Imaris software. In all movies, the cranial side of the embryo is on the left side.

**Zebrafish whole-mount in situ hybridization**. Whole-mount in situ hybridization was performed using antisense probes amplified from zebrafish embryonic cDNA, cloned into TOPO (Invitrogen, Carlsbad, USA), and linearized with corresponding restriction enzymes. Dig-labeled antisense RNA was synthesized using Roche Dig labeling kit (Roche, Mannheim, Germany).

**Isolation of pronephric tubules**. One- and two-day-old cdh17:GFP zebrafish embryos were anesthetized with 0.2% tricaine, and incubated in 10 mM DTT for 1 h at room temperature. The embryos were washed three times, and subsequently incubated at 28.5 °C in 5 mg/ml collagenase in Hank's solution with calcium for 3 h. The larvae were then passed through a pipette tip 4–5 times to disaggregate the tissue. The GFP-expressing pronephric tubules were collected under a dissecting microscope equipped with a fluorescent light source.

**RNA sequencing data analysis**. The quality trimming of raw RNA sequencing reads was performed using Trimmomatic[61]; remaining reads were aligned to the mouse genome version GRCm38 with subsequent calculation of read counts using STAR[62]. Genes with low expression values were filtered out, and raw read counts were normalized and transformed to log2-counts per million (logCPM)-values using edgeR library voom function from the limma package[63], respectively. Calibration for batch effects was done with the ComBat algorithm from the sva library[64]. GSEA was performed using the gage function[65] on GO-Terms and Consensus-Pathways. All GO-Terms and pathways were tested for regulation in a single direction (either up or down) using all possible combinations of knockout and control samples. All testing results were corrected for multiple testing using the method of Benjamini and Hochberg[66].

**NMR metabolomics**. All samples were recorded at 298 K on a Bruker Avance HD spectrometer operating at 700 MHz for the acquisition. The instrument was equipped with a 5 mm TCI cryoprobe with a Z-gradient. Maleic acid was used as internal standard for quantification and trimethylsilyl-3-propionic acid-d4 (TMSP) for the zero calibration. The samples were measured at 298 K. Urine samples (50 µl) were supplemented with 170 µl of deuterated phosphate buffer (DPB, pH 7.4), 30 µl of a 5 mM solution of maleic acid and 1 µl of a 10 mg/ml TMSP solution. 1H-NMR spectra were acquired using a 1D NOESY sequence with pre-saturation for urine samples. The Noesypresat experiment used a RD-90°-T1-90°-Tm-90°-acquire sequence with a relaxation delay of 4 s, a mixing time (Tm) of 10 ms, and a fixed T1 delay of 4 µs. Water-suppression pulse was placed during the relaxation delay (RD). The number of transient was 64 (64 K data points), and a number of 4 dummy scans was chosen. Acquisition time was fixed to 3.2769001 s. The data were processed with the Bruker Topspin 3.2 software with a standard parameter set. Phase and baseline corrections were performed manually over the entire range of the spectra, and the δ scale was calibrated to 0 ppm, using the internal standard TMSP. Gradient enhanced magnitude COSY experiment (pulse sequence cosygpprqf supplied by Bruker) with a pre-saturation during relaxation delay was used for 2D measurements. Spectra were collected with 4096 points in F2 and 300 points in F1 over a sweep width of 10 ppm, with six scans per F1 value. The acquisition times were fixed to 0.2557028 s in F2 and 0.0187212 in F1. The resulting COSY spectra were processed in Topspin 3.2 using standard methods, with sine-squared apodization in both dimensions and zero filling in F1 to yield a transformed 2D dataset of 2048 by 2048 points. Edited Hetero-nuclear single quantum correlation HSQC (pulse sequence HSQCEDETGP) with 256 increments, 32 transients, a 1.5 s relaxation delay, sweep width of 14 and 100 ppm, and offsets of 6.012 and 47 ppm was used. The data were processed on Topspin 3.2 using standard methods, with sine-squared apodization in both dimensions and zero filling in F1 to yield a transformed 2D dataset of 1024 by 1024 points.

**Multivariate analysis**. For statistical analysis, optimized 1H-NMR spectra were automatically baseline-corrected and reduced to ASCII files using AMIX software (version 3.9.14; Bruker). The spectral intensities were normalized to total intensities and reduced to integrated regions of equal width (0.04 ppm) corresponding to the 0.5–10.00 ppm region. Because of the residual signals of water and maleic acid, regions between 4.7–5 ppm (water signal) and 5.9–6.1 ppm (maleic acid signal) were removed before analysis. The reduced and normalized NMR spectral data were imported into SIMCA (version 13.0.3, Umetrics AB, Umea Sweden). Pareto scaling was applied to bucket tables, and discriminant analysis (DA) such as Principal Component Analysis (PCA) and Partial Least Squares Discriminant Analysis (PLS-DA) were performed. SIMCA was used to generate all PCA, PLS-DA models, and plots. PCA was used to detect possible outliers and determine intrinsic

clusters within the data set, while PLS-DA maximized the separation and facilitated the graphic visualization of differences and similarities between groups. The quality of PLS-DA models was determined by the goodness of fit ($R^2$); the predictability was calculated on the basis of the fraction correctly predicted in one-seventh cross-validation ($Q^2$).

**Metabolite identification**. From PLS-DA loading plots, metabolites with higher loadings were identified. Signals with values of Variable Importance in Projection (VIP) higher than 1 were considered as significant, and further validated using t-test with Metaboanalyst. Metabolite identification was next performed using the open-access database NMR suite 8.1 (Chenomx Inc., Edmonton, Canada), the free web-based tool HMDB (http://www.hmdb.ca) and tables. Each metabolite identified was finally confirmed by performing peak correlation plots from 2D-NMR spectra (COSY and HSQC).

**Pathway analysis**. The analyses of metabolic pathways were performed by Metaboanalyst (www.metaboanalyst.ca), using the Metabolic Set Enrichment Analysis (MSEA) with an Over Representation Analysis (ORA) algorithm.

**Measurement of extracellular acidification rates**. Tubular epithelial cells were isolated from 2-week old wild-type and Myc[fl/fl]*Pax8rtTA*TetOCre mice, and grown in DMEM-F12 medium. Kidneys were dissected and sliced into pieces of 1 mm. After digestion in collagenase, DNAse, and proteinase at 37 °C, renal fragments were sieved through two nylon sieves with a pore size of 100 µm, and cultured in DMEM-F12 containing 5% FCS, penicillin and streptomycin, epidermal growth factor (EGF), insulin, transferrin, selenit, dexamethason, L-thyroxin, and HEPES. For knockout induction doxycycline was added into culture medium for 24 h (0.5 µg/ml) followed by a 24-h incubation period without doxycycline. For metabolic profiling, the culture medium was replaced with glucose-free XF24 Seahorse medium, and cells were incubated without $CO_2$ at 37 °C for 1 h, followed by sequential addition of glucose (10 mM), oligomycin (4 µM), and 2-deoxy-D-glucose (50 mM) according to the recommendations of the manufacturer (Seahorse Bioscience). The ECAR values were normalized to the protein concentration in each well, determined at the end of the experiment.

## Data availability

The RNA sequencing data have been deposited in the GEO database under the accession code: GSE102044, the NMR data were deposited at MetaboLights (https://www.ebi.ac.uk/metabolights/). The authors declare that all data supporting the findings of this study are available within the article and its supplementary information files or from the corresponding author upon reasonable request.

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

## Acknowledgements

The authors gratefully acknowledge the technical assistance of Temel Kilic, Christina Engel, Barbara Müller, and Annette Schmitt, and members of the lab for critical input. We are very thankful to Darren Gilmore for generously providing multiple zebrafish lines. This work was supported by funds of the Deutsche Forschungsgemeinschaft (DFG) SFB 850 (M.B. and G.W.), SFB 1140 (F.G., T.B.H., and G.W.), the Excellence Initiative of the German Federal & State Governments (Grant EXC 294 BIOSS Centre for Biological Signalling Studies) (V.L. and G.W.), and the German Federal Ministry of Education and Research (BMBF) within the framework of the e:Med research and funding concept DeCaRe (M.B.).

## Author contributions

T.A.Y. and K.S. designed and performed the zebrafish experiments. A.P.T. designed and performed the mouse ischemia reperfusion experiments. J.W., M.G., A.S., I.H. and A.W. performed pronephros injury and cell tracking experiments. F.G. and T.B.H. designed the generation of the mouse lines and the ischemia reperfusion experiments. V.L. generated the *cxcr4b:H2bRFP* zebrafish line. A.T., T.B. and A.B. performed the Seahorse experiment. A.B. performed the Western blot analysis. F.J., J.L. and P.T. performed the metabolomics analysis. J.H. and M.B. carried out the data analysis of the mouse RNA seq data. A.K.Z. designed the zebrafish pronephros laser-injury model. G.W. conceived and designed the experiments, and wrote the paper.

## Additional information

**Competing interests:** The authors declare no competing interests.

