## [Peer Review File · Nature Communications]

Reviewers' Comments:

Reviewer #1:

Remarks to the Author:

The manuscript "cxcl12a/cxcr4b and myca signaling override collective cell migration to repair zebrafish pronephros injuries" by Yakulov et al report on the role of cxcl12/cxcr4 in regeneration of the pronephros in zebrafish. The topic of the paper is interesting at the basic biology and biomedical levels and in principle fits the scope of the journal. I am listing below comments by order of appearance in the paper (rather than importance), comments that I hope the authors find useful. The amount and nature of the comments, in my opinion suggest that the authors should conduct many more experiments to reach the level expected from a publication in nature communication.

1. The language should be improved at the grammar and spelling level. e.g. "Zebrafish myca was increased in cells participating in the repair response." → meaning the RNA level?, "requirement for cxc12a/cxcr4b signaling" → cxcl12a., "observed in cxcx12a/cxcr4b-deficient" → cxcl12a etc. (cxcl12 is misspelled in multiple positions).
2. There is a long discussion + background concerning pLLP migration. It could be that "historically", this is how the project evolved, but considering that the authors later claim that the mechanisms controlling the migration / the characteristics of the processes differ; these parts should be significantly shortened.
3. Similarly, the relatively detailed background concerning CXCR4 signaling prepares the reader for analysis at this level, which is not there – "CXCR4 is rapidly phosphorylated by G protein-coupled receptor kinases, followed by β-arrestin binding, recruitment of the E3 ubiquitin ligase AIP4, mono-ubiquitylation of carboxy-terminal lysine residues".
4. "while injured pronephric ducts before 30-36 hpf are repaired by the contraction of actomyosin bundles and a purse-string-like occlusion (Fig. 1f)" – The statement that the presence of actin and myosin at the occlusion site reflects "purse-string-like occlusion" would benefit from more detailed higher magnification presentation. In Fig 1f, indicate what the different colors mean within the panel.
5. The use of foxj1a and foxj1b morpholinos should be controlled by second morpholinos / phenotypic rescue, unless those morpholinos were described before in the same context.
6. "The absence of cxc12a or cxcr4 results in defective repair". A clear and detailed analysis of cxcr4 and cxcl12 expression in the pronephros at the relevant times should be presented, so the reader knows with no doubt where the signals originate and received. Also here cxcl12 is misspelled.
7. "induced a bidirectional migration of the neighboring cells to close the injured gap (Fig. 2). " – indicate how many times the experiment was performed, number of embryos etc.
8. Supplementary Movies 1-3 are nice, but it is not exactly clear to me what the double injury teaches that the single cut cannot.
9. "we compared the expression profile of micro-dissected one and two day-old zebrafish pronephric tubules" – describe procedure in detail.
10. "Using the ZFIN database..." not clear how it was used.
11. The differences in speed the authors claim exist should be presented with specific values, with the number of cells and embryos the data is derived from. e.g. in Figure 2C – is displacement = speed? How many cells were used to derive the data etc.
12. "revealed an up-regulation of cxcr4b expression in tubule cells adjacent to the injury.." From the movie it seems like upregulation occurs over the whole posterior duct. Is that meaningful?
13. "The cxcr4b/cxcl12a signaling module is not required for normal zebrafish pronephros development" I would suggest presenting this information before the cxcr4 phenotypes are presented. More critically, this point should be examined also at early stages (times when cutting experiments were conducted) to exclude defects at those times as well. Otherwise, one could argue that there are early defects that are manifested as regeneration defects at later stages. Concerning this section, it would be interesting to compare the speed of cells of different genotypes in regeneration (or lack of it) at the different stages.

14. The data in Supplementary figure 2d is not clear and there is a need for number of repeats etc. the effect does not seem very local and although a different orientation, the fish in d appears to have less signal.
15. In 4c and d- indicate number of embryos checked. Why wasn't a rescue experiment conducted in figure 4c? The rescue experiments should be controlled by injection of an inactive RNA.
16. "suggesting that myca is required to transiently override the posterior-to-anterior cell migration" - is the elevation in cxcr4b in response to injury affected by the myc morpholino?
17. The authors conclude that cxcr4/cxcl12 signaling is required for the repair at a late stage and based of the expression pattern of the genes conclude that the time of action is the time of repair. Nevertheless, the option that some early defects are manifested late was not ruled out. A good experiment that would support the model the authors suggest would be to provide the cxcr4 just before the time of injury (e.g. using a heat shock promoter, drug-induced expression or specific rescue in the duct). If such an experiment is "successful" it supports the claims of the authors. Conducting a similar experiment with cxcl12 would address the point of whether the chemokine actually directs the migration.
18. Concerning the mouse experiments the authors should present or cite the detailed expression pattern of cxcl12 and cxcr4 in the tissue at the relevant times.
19. The mouse experiments were conducted after the doxycycline induction. Could it be that the results obtained reflect early defects that were manifested later i.e. at the time the experiments are conducted? The "integrity/proper state" of the tissue should be confirmed before the experiment is conducted.
20. "...knockout mice at this time after injury (Fig. 5d, and Supplementary Fig. 10).." – mark the magnified panels in 5d and h with numbers not letters, so there is no confusion with the panel labeling. One way or the other, it is not clear what one should observe in those panels, an issue that should be explained better.
21. "...revealed increased Cxcr4 expression in both Cxcl12 and Myc KO..". Isn't it the opposite of the zebrafish findings? This issue should be explained. Regarding the "CXCR4 signaling is up-regulated to compensate for the loss of Cxcl12" – how do the authors envision a compensation for lack of ligand by up-regulation of the receptor?
22. The statement "This difference suggests that cxcl12a/cxcr4b signaling is not required for the slow mode of migration.." is a bit strong concerning that there are only two examples checked.
23. In the methods part, provide more details concerning the way the 2-photon ablation experiments were conducted.

Reviewer #2:

Remarks to the Author:

The authors have done a very professional job in studying the zebrafish pronephros following laser induced injury. They find that the injury-induced gap is closed by migration of cells from both ends, a process that seems to be dependent on CXCL12/CXCR4 function as well as that of Myca. whether that is an overriding of collective cell migration is not really made clear. Presumably the migrating cells of the pronephros are already surrounded by an extracellular matrix sheath whereas classic collective cells migration (as occurs in the lateral line occurs by groups of cells not already "trapped" in a pre-formed tubule.

They then induced ischemic injury in mouse kidneys and find that deletion of CXCL12 in Pax8 nephrons (i.e. most of the kidney) increases BUN, but really does not seem to have any worsening of the histological damage. Although the studies suggest that CXCL12 does play some role in AKI, the work is not detailed enough to tell us what exactly this role is. There have been claims in the literature that ischemic damage kills some proximal tubule cells which are shed and replaced by division of neighboring cells. There is hardly any migration that happens there; it is the next door cell that replaces it. So the relationship of this process to what is observed in the pronephros is pretty obscure. At any rate, no evidence is provided that this replacement of neighboring cells is prevented by CXCL12 deletion.

The juxtaposing of the two models of renal injury seems to me to be setting a contrived type of

analogy. There is no reason to think that ischemia (which activates HIF (the primary regulator of CXCL12/CXCR4) is similar to laser induced cell killing. Further, AKI induced by ischemia seems to involve a variety of other invading immune cells which contribute to the cell damage. Finally, the histological findings do not seem to show any difference between knockout and WT mice. Thus although the two parts of the paper are well done and I have no technical issues with either parts, I feel that the findings are simply of "archival" interest rather than one that sheds light on either mammalian AKI or zebrafish pronephros development.

Reviewers' comments:

Reviewer #1 (Remarks to the Author):

The manuscript “cxcl12a/cxcr4b and myca signaling override collective cell migration to repair zebrafish pronephros injuries” by Yakulov et al report on the role of cxcl12/cxcr4 in regeneration of the pronephros in zebrafish. The topic of the paper is interesting at the basic biology and biomedical levels and in principle fits the scope of the journal. I am listing below comments by order of appearance in the paper (rather than importance), comments that I hope the authors find useful. The amount and nature of the comments, in my opinion suggest that the authors should conduct many more experiments to reach the level expected from a publication in nature communication.

1. The language should be improved at the grammar and spelling level. e.g. “Zebrafish myca was increased in cells participating in the repair response.” → meaning the RNA level?, “requirement for cxc12a/cxcr4b signaling” → cxcl12a., “observed in cxcx12a/cxcr4b-deficient”-> cxcl12a etc. (cxcl12 is misspelled in multiple positions).

The spelling and grammatical errors were corrected. Thank you for the careful reading of the manuscript.

2. There is a long discussion + background concerning pLLP migration. It could be that “historically”, this is how the project evolved, but considering that the authors later claim that the mechanisms controlling the migration / the characteristics of the processes differ; these parts should be significantly shortened.

This part was omitted from the introduction.

3. Similarly, the relatively detailed background concerning CXCR4 signaling prepares the reader for analysis at this level, which is not there – “CXCR4 is rapidly phosphorylated by G protein-coupled receptor kinases, followed by β -arrestin binding, recruitment of the E3 ubiquitin ligase AIP4, mono-ubiquitylation of carboxy-terminal lysine residues”.

This part was omitted from the introduction.

4. “while injured pronephric ducts before 30-36 hpf are repaired by the contraction of actomyosin bundles and a purse-string-like occlusion (Fig. 1f)” – The statement that the presence of actin and myosin at the occlusion site reflects “purse-string-like occlusion” would benefit from more detailed higher magnification presentation. In Fig 1f, indicate what the different colors mean within the panel.

The schematic of the previous submission (Fig. 1f) is replaced by a single confocal plane, which quantifies the accumulation of actomyosin. The histogram depicts levels of phosphorylated myosin, with two peaks adjacent to the injury, supporting an apical constriction of the tubular duct cells.

5. The use of *foxj1a* and *foxj1b* morpholinos should be controlled by second morpholinos / phenotypic rescue, unless those morpholinos were described before in the same context.

The utilized *foxj1a* and *foxj1b* MOs were previously described (Hellman et al., 2010). The reference is cited in the corresponding supplemental figure legend.

6. “The absence of *cxc12a* or *cxcr4* results in defective repair”. A clear and detailed analysis of *cxcr4* and *cxcl12* expression in the pronephros at the relevant times should be presented, so the reader knows with no doubt where the signals originate and received. Also here *cxcl12* is misspelled.

Both *cxcl12a* and *cxcr4b* are expressed in the zebrafish pronephros. Expression of *cxcl12/sdf1* in the zebrafish pronephros at 2 dpf is published (David et al., 2002). This reference was now included.

Expression of *Cxcr4b* is presented in Supplementary Fig. 7, using the *cxcr4b:cxcr4b-tFT* zebrafish line. Note that *cxcr4b* is strongly expressed in the corpuscle of Stannius. Up-regulation of *cxcr4b* is demonstrated by *in situ* hybridization in Fig. 3j, and by time-lapse video-microscopy in Supplementary Movies 6,7, using the *Cxcr4b:H2B-RFP* reporter.

Cxcr4b protein is rapidly degraded. However, the non-degradable *Cxcr4b*-mem-tFT clearly accumulates in the pronephros, supporting the expression of *cxcr4b* in the zebrafish pronephros (Supplemental Fig. 7d,e,f). *Cxcr4b* is up-regulated in cells adjacent to the injury as demonstrated by *in situ* hybridization in Figure 3g, g’.

To demonstrate the importance of *Cxcl12a*-mediated directed cell migration, we performed injuries in the presence of ectopic *Cxcl12a*, using the heat-shock inducible zebrafish line *hsp70:cxcl12a*. As shown in Supplemental Fig. 7g, heat shock and ectopic expression of *Cxcl12a* increased the number of pronephric tubules that were not repaired after injury. These results are consistent with the arrested migration of the pLLP after ectopic *Cxcl12* expression, and provide further evidence that the *Cxcl12a/Cxcr4b* module plays an important role in the repair process after laser-induced ablation.

7. “induced a bidirectional migration of the neighboring cells to close the injured gap (Fig. 2).
“ – indicate how many times the experiment was performed, number of embryos etc.

Laser-mediated pronephros injuries were performed in over 1,000 zebrafish embryos. For the initial set of experiments presented in Figure 2, we performed laser-mediated pronephros injuries in over 250 embryos. After examination under low magnification fluorescent stereomicroscope, 90 were further analyzed by WISH with various antisense probes. The complete healing process was observed in 43 control and experimental conditions using high-resolution confocal time-lapse movies. Seven representative movies were selected for detailed cell tracking using the Imaris software.

We also analyzed the regeneration efficiency in wild-type zebrafish embryos with wounds of different sizes: two days after fertilization, a 50- μ m injury, equivalent to approximately 5-6 cell diameters, was repaired in 100% (n=14), a 100- μ m injury (equivalent to 10-12 cell diameters) was repaired in 80% (n=20), and none of the 150- μ m gaps was repaired (n=17). This data is now presented in Supplementary Fig. 4d.

8. Supplementary Movies 1-3 are nice, but it is not exactly clear to me what the double injury teaches that the single cut cannot.

The double injury model demonstrates that the cell cluster isolated by two injuries reverses cell migration at the posterior wound while maintaining the direction at the anterior side. In the absence of cell proliferation, the double injury results in visible flattening of the tubular epithelial cells in the isolated patch. Thus, the migratory response overrides not only the direction of the posterior-to-anterior cell migration but also cellular programs that normally maintain cell shape.

9. “we compared the expression profile of micro-dissected one and two day-old zebrafish pronephric tubules” – describe procedure in detail.

The isolation of pronephric tubules is now described in the Method section.

10. “Using the ZFIN database...” not clear how it was used.

We used the ZFIN database to identify candidate genes previously identified in the zebrafish pronephros by *in situ* hybridization. A more detailed description of our approach is included in the revised manuscript.

11. The differences in speed the authors claim exist should be presented with specific values, with the number of cells and embryos the data is derived from. e.g. in Figure 2C – is displacement = speed? How many cells were used to derive the data etc.

We repeated the experiments measuring both displacement length (i.e. the net distance traveled by cell) and the mean track speed in Figure 2c. Cell number and number of experiments are now included in Figure 2c and the corresponding figure legend.

12. “revealed an up-regulation of *cxcr4b* expression in tubule cells adjacent to the injury..” From the movie it seems like upregulation occurs over the whole posterior duct. Is that meaningful?

Both *in situ* hybridization and the time-lapse movies show an up-regulation of *Cxcr4b* in the tubular epithelial cells immediately surrounding the site of the injury. There is also strong accumulation of *Cxcr4b* in the corpuscle of Stannius. The *Cxcr4b* up-regulation in more distal cells likely represents a secondary, stretch-activated expression of *cxcr4b* elicited by the faster migration of the cells next to the injury (Li et al., 2009; Zhou et al., 2013).

13. “The *cxcr4b/cxcl12a* signaling module is not required for normal zebrafish pronephros development” I would suggest presenting this information before the *cxcr4* phenotypes are presented. More critically, this point should be examined also at early stages (times when cutting experiments were conducted) to exclude defects at those times as well. Otherwise, one could argue that there are early defects that are manifested as regeneration defects at later stages. Concerning this section, it would be interesting to compare the speed of cells of different genotypes in regeneration (or lack of it) at the different stages.

As suggested by the reviewer, we start with data depicting the track speed and displacement in control and mutant zebrafish embryos (Figure 3a,b and Supplementary Fig. 6).

We repeated these experiments, determining the mean track speed and track displacement at 36 hpf, 41 hpf and 48 hpf in wild-type (control) and mutant zebrafish lines. As shown in Figure 3 b, and Supplementary Fig. 6, there are no differences at the three different time points relevant for the recordings after laser-induced injuries. Note that Supplementary Fig. 6 represents the complete data set, including the results of Figure 3b.

14. The data in Supplementary figure 2d is not clear and there is a need for number of repeats etc. the effect does not seem very local and although a different orientation, the fish in d appears to have less signal.

There is no Supplementary Fig. 2d. Supplementary Fig. 3d depicts a normal repair response despite knockdown of *foxj1*. We have now added the numbers of repeats to both Supplementary Fig. 2 and Supplementary Fig 3d.

15. In 4c and d- indicate number of embryos checked. Why wasn't a rescue experiment conducted in figure 4c? The rescue experiments should be controlled by injection of an inactive RNA.

The rescue in Figure 4c was performed as requested. The rescue experiments were controlled by injection of an inactive RNA (*GFP mRNA*, *myca mRNA* containing four nucleotide substitutions), which in contrast to the active mRNA did not significantly rescue the repair defect. The results are depicted in Supplementary Fig. 9c,d.

16. "suggesting that *myca* is required to transiently override the posterior-to-anterior cell migration" - is the elevation in *cxcr4b* in response to injury affected by the *myc* morpholino?

In situ hybridization for *cxcr4b* did not reveal significant differences after knockdown *myca* indicating that the injury-mediated increase of *cxcr4b* expression is not only mediated by *myca*. The results are depicted in Supplementary Fig. 10d. The splice-blocking morpholino oligonucleotide was used to deplete zebrafish *myca*. Zebrafish embryos were injured 2 dpf, followed by *in situ* hybridization to detect *cxcr4b* mRNA in the tubular epithelial cells adjacent to the injury.

17. The authors conclude that *cxcr4/cxcl12* signaling is required for the repair at a late stage and based of the expression pattern of the genes conclude that the time of action is the time of repair. Nevertheless, the option that some early defects are manifested late was not ruled out. A good experiment that would support the model the authors suggest would be to provide the *cxcr4* just before the time of injury (e.g. using a heat shock promoter, drug-induced expression or specific rescue in the duct). If such an experiment is "successful" it supports the claims of the authors. Conducting a similar experiment with *cxcl12* would address the point of whether the chemokine actually directs the migration.

Following the advice of the reviewer, we used the *hsp70:cxcl12a* transgenic zebrafish line. Heat shock, triggering ubiquitous expression of *Cxcl12a*, resulted in a strong repair defect. This finding supports the hypothesis that the *Cxcl12a/Cxcr4b* module represents an essential component of the repair process in response to injury. The data is presented in Supplementary Fig. 7g.

18. Concerning the mouse experiments the authors should present or cite the detailed expression pattern of *cxcl12* and *cxcr4* in the tissue at the relevant times.

Several authors have demonstrated expression of *CXCL12/CXCR4* and/or up-regulation *CXCL12/CXCR4* after acute kidney injury (e.g. (Ge et al., 2017; Liu et al., 2012; Mazzinghi et al., 2008; Ohnishi et al., 2015; Oliver et al., 2012; Stokman et al., 2010; Togel et al., 2005)). These citations were included in the revised manuscript. Most authors have interpreted the expression of *CXCL12* as signal that attracts bone marrow-derived cells to the injured kidney. However, several recent publications argue against renal recruitment of bone-marrow cells, but rather suggest that neighboring tubular epithelial cells repair injuries after ischemic damage.

19. The mouse experiments were conducted after the doxycycline induction. Could it be that the results obtained reflect early defects that were manifested later i.e. at the time the experiments are conducted? The “integrity/proper state” of the tissue should be confirmed before the experiment is conducted.

We measured urea concentrations and performed tissue sections in control and knockout mice immediately before ischemia/reperfusion injury. As shown in Supplementary Fig. 11 and Supplementary Fig. 13, there were no differences between control and knockout animals before the ischemia/reperfusion experiment.

20. “...knockout mice at this time after injury (Fig. 5d, and Supplementary Fig. 10)..” – mark the magnified panels in 5d and h with numbers not letters, so there is no confusion with the panel labeling. One way or the other, it is not clear what one should observe in those panels, an issue that should be explained better.

The labeling was changed accordingly in Fig. 5d and Fig. 5h, and the corresponding Fig. 10 and Fig. 11. We also changed Fig. 6c accordingly, and used numbers instead of letters.

It is important to demonstrate by histology and the corresponding magnifications that both control and genetically modified (i.e. the *Cxcl12*- and *Myc*-deficient) animals suffered severe ischemia reperfusion injuries, characterized by necrotic tubular epithelial cells and the accumulation of obstructing casts within the tubular lumen. Although there were no obvious differences detectable by histology, the renal failure in *Cxcl12*- and *Myc*-deficient animals was aggravated as determined by BUN and NGAL.

The lack of structural differences suggests that the different outcome is the result of early adaptive changes such as cell migration and/or metabolic responses that are not detectable by histology.

21. “..revealed increased *Cxcr4* expression in both *Cxcl12* and *Myc* KO..”. Isn’t it the opposite of the zebrafish findings? This issue should be explained. Regarding the “CXCR4 signaling is up-regulated to compensate for the loss of *Cxcl12*” – how do the authors envision a compensation for lack of ligand by up-regulation of the receptor?

Cxcr4 is not only controlled by MYC, but other factors that apparently compensate for the loss of MYC, and up-regulate *Cxcr4* in response to ischemia/reperfusion (I/R) injury. The CXCR4 promoter includes four hypoxia-response elements located within 2.6 kp upstream of the transcriptional start site, and is up-regulated by HIF-1 α in response to hypoxia (Staller et al., 2003).

We used laser-mediated ablation of renal tubular epithelial cells in zebrafish embryos to identify molecules involved in early repair. It is unlikely that this model induces ischemia-typical mediated responses. However, the zebrafish model successfully identified CXCL12 and MYC as components that maintain crucial functions immediately after I/R injury, suggesting that the zebrafish and mouse models share fundamental programs that are initiated after epithelial cell necrosis. The PAX8-driven excision of CXCL12 eliminates CXCL12 from tubular epithelial cells, but does not prevent CXCL12 expression in surrounding tissues or infiltrating cells. However, even the limited elimination of CXCL12 aggravated renal failure after I/R injury, and resulted in a distinct transcriptional signature.

22. The statement “This difference suggests that *cxcl12a/cxcr4b* signaling is not required for the slow mode of migration..” is a bit strong concerning that there are only two examples checked.

We have modified the statement accordingly.

23. In the methods part, provide more details concerning the way the 2-photon ablation experiments were conducted.

We have included a reference that provides a detailed description of the laser-ablation method used in this manuscript (Johnson et al., 2011).

Reviewer #2 (Remarks to the Author):

The authors have done a very professional job in studying the zebrafish pronephros following laser induced injury. They find that the injury-induced gap is closed by migration of cells from both ends, a process that seems to be dependent on CXCL12/CXCR4 function as well as that of Myca. whether that is an overriding of collective cell migration is not really made clear. Presumably the migrating cells of the pronephros are already surrounded by an extracellular matrix sheath whereas classic collective cell migration (as occurs in the lateral line occurs by groups of cells not already "trapped" in a pre-formed tubule.

The proximal segment of the zebrafish pronephros is shaped by cell movements that have been characterized as "collective cell migration" by Iain Drummond's group (Vasilyev et al., 2009). We agree with the reviewer that there are substantial differences between the collective cell migration of the zebrafish pLLP and the pronephros. However, it was surprising that the chemokine signaling pathway, while essential for directed migration of the pLLP, is essential only after a significant disruption to the integrity of the pronephros. We identified rapid migration as a key factor differentiating the concerted movement of these two tissues: pLLP cells migrate at a speed nearly 10x higher than pronephros cells. Additional data strengthens the hypothesis that Cxcl12a/Cxcr4b signaling is required to mediate fast cell migration in response to injury.

They then induced ischemic injury in mouse kidneys and find that deletion of CXCL12 in Pax8 nephrons (i.e. most of the kidney) increases BUN, but really does not seem to have any worsening of the histological damage. Although the studies suggest that CXCL12 does play some role in AKI, the work is not detailed enough to tell us what exactly this role is. There have been claims in the literature that ischemic damage kills some proximal tubule cells which are shed and replaced by division of neighboring cells. There is hardly any migration that happens there; it is the next door cell that replaces it. So the relationship of this process to what is observed in the pronephros is pretty obscure. At any rate, no evidence is provided that this replacement of neighboring cells is prevented by CXCL12 deletion.

Most renal ischemia/reperfusion (I/R) models focus on functional and morphological changes occurring \geq 48-96 hours after the initial injury. To avoid repair responses mediated by cell proliferation, we selected an early time point (12 hr) after I/R injury to determine the functional impact of CXCL12 and MYC deficiency before the onset of proliferation. As the reviewer points out, it is speculated that necrotic cells are initially replaced by neighboring cells before a proliferative burst replaces lost cells. Our data support the hypothesis that the early adaptive response involves cell migration. It is unsurprising that histology reveals minimal changes if the aggravated renal failure is caused by defective cell migration.

The juxtaposing of the two models of renal injury seems to me to be setting a contrived type

of analogy. There is no reason to think that ischemia (which activates HIF (the primary regulator of CXCL12/CXCR4) is similar to laser induced cell killing.

Laser-mediated ablation of epithelial cells and ischemia-induced kidney injury represent two very different etiologies. The zebrafish model allowed us to investigate immediate molecular repair responses after injury which are inaccessible in a mammalian model. After differentiating two types of repair processes and identifying the *Cxcl12/Cxcr4b* module and *Myca* in the zebrafish model, we showed that these molecules are involved in ischemia/reperfusion injury in the mammalian kidney. The zebrafish model suggests that these molecules participate in a migratory repair response; in knockout mice, RNA sequencing revealed common abnormalities in mitochondrial metabolism after elimination of either *Cxcl12* or *Myc*. Analysis of urine metabolites revealed that both molecules support glycolysis during acute kidney injury. Interestingly, laser-induced cell ablation induces *cxcr4* expression, thus resembling ischemia/HIF-induced *Cxcl12/Cxcr4b* expression.

Further, AKI induced by ischemia seems to involve a variety of other invading immune cells which contribute to the cell damage. Finally, the histological findings do not seem to show any difference between knockout and WT mice.

Inflammatory reaction in response to IR injury plays an important role in mammalian AKI. However, these reactions occur at later stages. An early time point after injury was selected to avoid an impact of invading cells on kidney function. We were primarily interested to identify immediate responses after injury (using the zebrafish pronephros as a model system), and validating the results in mice with a kidney-specific, inducible knockout.

Thus although the two parts of the paper are well done and I have no technical issues with either parts, I feel that the findings are simply of "archival" interest rather than one that sheds light on either mammalian AKI or zebrafish pronephros development.

Progress in fundamental understanding and clinical treatment of acute kidney injury has frustratingly limited. Mammalian AKI has been characterized more on a cellular than a molecular basis, and not during early events. Several laboratories including those of Bonventre and Humphreys have made significant contributions by identifying the cells involved in the repair process after injury, refuting the idea that extra-renal stem cells replace necrotic tubular epithelial cells. However, molecular mechanisms underlying early adaptation and repair have remained largely elusive. Due to technical and methodological issues, it is difficult to elucidate these events in mammalian models of acute kidney injury *in vivo*. Consequently, it is currently unknown how the kidney reacts to an injury (ischemic and/or toxic) over the first initial hours. This limitation has hampered further insight into the immediate molecular adaptive mechanisms employed in response to injury, and thereby hampered the development of a targeted rational therapeutic strategy. Treatment of acute kidney injury continues to be largely supportive. Despite the increased sophistication of these supportive measures, morbidity and mortality rates due to AKI have remained unchanged over the last couple of decades.

Laser-mediated zebrafish pronephros injury is not equivalent to acute kidney injury in mammals, which caused by a combination of toxic and ischemic insults. Therefore, we validated key findings, i.e. the up-regulation of CXCL12/CXCR4 and MYC in mice with a kidney-specific, tetracycline-inducible knockout. The mouse ischemia/reperfusion injury model is considered analogous to human acute kidney injury. Using this well-established protocol, we observed that the lack of either CXCL12 or MYC aggravated the renal failure in mice exposed to ischemia/reperfusion.

Both molecules have been implicated in cell migration as well as tumor invasion, consistent with the defective migratory response that we observed after a laser-induced injury in zebrafish embryos lacking either molecule. Our in-depth analysis of these two molecules in the mouse model, however, revealed surprising new insights into the mechanisms that support the repair process. First, both molecules support mitochondrial homeostasis, and the altered retinoic acid metabolism uncovered in the knockout mice was ameliorated by tretinoin, demonstrating that linking zebrafish to mouse models results in new strategies to treat acute kidney injury. Second, the NMR-based metabolic analysis revealed that the deficiency of either *Myc* or *Cxcl12* compromises glycolysis in response to ischemia, signified by the lack of lactate in the urine of the knockout animals. That both knockout models display the same urinary alterations in response to injury highlights the importance of a high glycolytic flux in response to ischemic injury. Our observation may not only lead to new strategies to support renal tubular adaptation in response to ischemia, but also suggest that depriving tubular epithelial cells of glucose (e.g. through inhibition of SGLT2) may render kidney cells particularly susceptible to ischemia injury.

In summary, the combination of two very different models of acute kidney injury has revealed profound insight into the mechanisms that are important during early adaptation in response to an injury (laser or ischemic), and demonstrate that screening for candidates involved in the repair process in zebrafish and subsequent validation in a mammalian kidney injury model represents a powerful strategy to elucidate and treat acute kidney injury, a complication of severe human disease, where progress has been lacking for several decades.

References

- Cliff, T.S., Wu, T., Boward, B.R., Yin, A., Yin, H., Glushka, J.N., Prestegard, J.H., and Dalton, S. (2017). MYC Controls Human Pluripotent Stem Cell Fate Decisions through Regulation of Metabolic Flux. *Cell Stem Cell* 21, 502-516 e509.
- Corbet, C., and Feron, O. (2017). Cancer cell metabolism and mitochondria: Nutrient plasticity for TCA cycle fueling. *Biochim Biophys Acta* 1868, 7-15.
- David, N.B., Sapede, D., Saint-Etienne, L., Thisse, C., Thisse, B., Dambly-Chaudiere, C., Rosa, F.M., and Ghysen, A. (2002). Molecular basis of cell migration in the fish lateral line: role of the chemokine receptor CXCR4 and of its ligand, SDF1. *Proc Natl Acad Sci U S A* 99, 16297-16302.
- Draoui, N., and Feron, O. (2011). Lactate shuttles at a glance: from physiological paradigms to anti-cancer treatments. *Dis Model Mech* 4, 727-732.
- Ge, G., Zhang, H., Li, R., and Liu, H. (2017). The Function of SDF-1-CXCR4 Axis in SP Cells-Mediated Protective Role for Renal Ischemia/Reperfusion Injury by SHH/GLI1-ABCG2 Pathway. *Shock* 47, 251-259.
- Guo, F., Wang, Y., Liu, J., Mok, S.C., Xue, F., and Zhang, W. (2016). CXCL12/CXCR4: a symbiotic bridge linking cancer cells and their stromal neighbors in oncogenic communication networks. *Oncogene* 35, 816-826.
- Hellman, N.E., Liu, Y., Merkel, E., Austin, C., Le Corre, S., Beier, D.R., Sun, Z., Sharma, N., Yoder, B.K., and Drummond, I.A. (2010). The zebrafish *foxj1a* transcription factor regulates cilia function in response to injury and epithelial stretch. *Proc Natl Acad Sci U S A* 107, 18499-18504.
- Johnson, C.S., Holzemer, N.F., and Wingert, R.A. (2011). Laser ablation of the zebrafish pronephros to study renal epithelial regeneration. *J Vis Exp*.
- Kim, J.W., Zeller, K.I., Wang, Y., Jegga, A.G., Aronow, B.J., O'Donnell, K.A., and Dang, C.V. (2004). Evaluation of myc E-box phylogenetic footprints in glycolytic genes by chromatin immunoprecipitation assays. *Mol Cell Biol* 24, 5923-5936.

- Li, F., Guo, W.Y., Li, W.J., Zhang, D.X., Lv, A.L., Luan, R.H., Liu, B., and Wang, H.C. (2009). Cyclic stretch upregulates SDF-1alpha/CXCR4 axis in human saphenous vein smooth muscle cells. *Biochem Biophys Res Commun* 386, 247-251.
- Li, F., Wang, Y., Zeller, K.I., Potter, J.J., Wonsey, D.R., O'Donnell, K.A., Kim, J.W., Yustein, J.T., Lee, L.A., and Dang, C.V. (2005). Myc stimulates nuclearly encoded mitochondrial genes and mitochondrial biogenesis. *Mol Cell Biol* 25, 6225-6234.
- Liu, H., Liu, S., Li, Y., Wang, X., Xue, W., Ge, G., and Luo, X. (2012). The role of SDF-1-CXCR4/CXCR7 axis in the therapeutic effects of hypoxia-preconditioned mesenchymal stem cells for renal ischemia/reperfusion injury. *PLoS One* 7, e34608.
- Mazzinghi, B., Ronconi, E., Lazzeri, E., Sagrinati, C., Ballerini, L., Angelotti, M.L., Parente, E., Mancina, R., Netti, G.S., Becherucci, F., *et al.* (2008). Essential but differential role for CXCR4 and CXCR7 in the therapeutic homing of human renal progenitor cells. *J Exp Med* 205, 479-490.
- Mercurio, L., Cecchetti, S., Ricci, A., Pacella, A., Cigliana, G., Bozzuto, G., Podo, F., Iorio, E., and Carpinelli, G. (2017). Phosphatidylcholine-specific phospholipase C inhibition downregulates CXCR4 expression and interferes with proliferation, invasion and glycolysis in glioma cells. *PLoS One* 12, e0176108.
- Ohnishi, H., Mizuno, S., Mizuno-Horikawa, Y., and Kato, T. (2015). Stromal cell-derived factor-1 (SDF1)-dependent recruitment of bone marrow-derived renal endothelium-like cells in a mouse model of acute kidney injury. *J Vet Med Sci* 77, 313-319.
- Oliver, J.A., Maarouf, O., Cheema, F.H., Liu, C., Zhang, Q.Y., Kraus, C., Zeeshan Afzal, M., Firdous, M., Klinakis, A., Efstratiadis, A., *et al.* (2012). SDF-1 activates papillary label-retaining cells during kidney repair from injury. *Am J Physiol Renal Physiol* 302, F1362-1373.
- Staller, P., Sulitkova, J., Lisztwan, J., Moch, H., Oakeley, E.J., and Krek, W. (2003). Chemokine receptor CXCR4 downregulated by von Hippel-Lindau tumour suppressor pVHL. *Nature* 425, 307-311.
- Stokman, G., Stroo, I., Claessen, N., Teske, G.J., Florquin, S., and Leemans, J.C. (2010). SDF-1 provides morphological and functional protection against renal ischaemia/reperfusion injury. *Nephrol Dial Transplant* 25, 3852-3859.
- Togel, F., Isaac, J., Hu, Z., Weiss, K., and Westenfelder, C. (2005). Renal SDF-1 signals mobilization and homing of CXCR4-positive cells to the kidney after ischemic injury. *Kidney Int* 67, 1772-1784.
- Vasilyev, A., Liu, Y., Mudumana, S., Mangos, S., Lam, P.Y., Majumdar, A., Zhao, J., Poon, K.L., Kondrychyn, I., Korzh, V., *et al.* (2009). Collective Cell Migration Drives Morphogenesis of the Kidney Nephron. *PLoS Biol* 7, e9.
- Zager, R.A., Johnson, A.C., and Becker, K. (2014). Renal cortical pyruvate depletion during AKI. *J Am Soc Nephrol* 25, 998-1012.
- Zhou, S.B., Wang, J., Chiang, C.A., Sheng, L.L., and Li, Q.F. (2013). Mechanical stretch upregulates SDF-1alpha in skin tissue and induces migration of circulating bone marrow-derived stem cells into the expanded skin. *Stem Cells* 31, 2703-2713.

Reviewers' Comments:

Reviewer #1:

Remarks to the Author:

In the revised version of the manuscript the authors improved a few aspects and addressed some of the criticism raised.

Nevertheless, due to the following issues, I consider the paper not suitable for publication in its current form.

List of issues not addressed-

Point 5 –

The authors do not provide the sequence of the morpholino they used. The reference the authors provide for the morpholinos does not deal directly with the same process (regeneration in response to ablation) and in any case, also in the old paper cited the specificity of the morpholinos was not rigorously checked. Considering this, it is not clear why the authors chose not to try other set of morpholinos to substantiate their claims

One way or the other, as a result of the very large number of artifacts stemming from non-specific effects of morpholinos, the authors should follow the regulations in place concerning this reagent (Stainier et al 2017 (<https://doi.org/10.1371/journal.pgen.1007000>)). As mentioned before, a second pair of morpholinos should be employed as well as generation of genetic mutant fish, or use of Cas9/CRISPR – mediated knockout in G0 injections and phenotypic analysis.

Point 6 –

The authors do not present the expression pattern of cxcl12a in the relevant tissue and at the right time as requested. The reference they cite shows embryos at an earlier stage than the time they analyze the process.

Importantly, based on the findings of Venkiteswaran 2013 (Fig1C <https://doi.org/10.1016/j.cell.2013.09.046>) cxcl12a RNA is actually not expressed in the region of the pronephric duct in 36hpf zebrafish embryos.

Point 17 –

The criticism here relates to the option that the defects in regeneration observed in the cxcl12a / cxcr4b mutants result from defects related to earlier function of the proteins rather than from a direct role in the process.

Here the authors should have provided Cxcr4b until the time of the experiment and should have stopped it just before injury induction (e.g. around 32 hpf). This was not done and the ability to “confuse” the process by Cxcl12a expression does not prove that the chemokine functions in the process in a positive direction.

Point 21 –

The answer of the authors concerning the question “CXCR4 signaling is up-regulated to compensate for the loss of Cxcl12 – how do the authors envision a compensation for lack of ligand by up-regulation of the receptor” is that “The PAX8-driven excision of CXCL12 eliminates CXCL12 from tubular epithelial cells, but does not prevent CXCL12 expression in surrounding tissues or infiltrating cells.”. This answer is not supported by any data concerning the expression of Cxcl12 in neighboring tissues.

Reviewer #2:

Remarks to the Author:

I did not find the response to my comments convincing.

Response to the reviewers' comments

Reviewer #1 (Remarks to the Author):

In the revised version of the manuscript the authors improved a few aspects and addressed some of the criticism raised.

Nevertheless, due to the following issues, I consider the paper not suitable for publication in its current form.

List of issues not addressed-

Point 5 –

The authors do not provide the sequence of the morpholino they used. The reference the authors provide for the morpholinos does not deal directly with the same process (regeneration in response to ablation) and in any case, also in the old paper cited the specificity of the morpholinos was not rigorously checked. Considering this, it is not clear why the authors chose not to try other set of morpholinos to substantiate their claims. One way or the other, as a result of the very large number of artifacts stemming from non-specific effects of morpholinos, the authors should follow the regulations in place concerning this reagent (Stainier et al 2017 (<https://doi.org/10.1371/journal.pgen.1007000>)). As mentioned before, a second pair of morpholinos should be employed as well as generation of genetic mutant fish, or use of Cas9/CRISPR – mediated knockout in G0 injections and phenotypic analysis.

Point #5 and our response to the first review was as follows:

5. The use of *foxj1a* and *foxj1b* morpholinos should be controlled by second morpholinos / phenotypic rescue, unless those morpholinos were described before in the same context.

The utilized *foxj1a* and *foxj1b* MOs were previously described (Hellman et al., 2010). The reference is cited in the corresponding supplemental figure legend.

In the cited article (Hellman et al., *PNAS* 2010), Drummond's group carefully delineated the role of *foxj1* as a stretch-activated transcription factor, using ATG- and splice-blocking morpholino oligonucleotides (MOs). Experiments were performed between 2 and 3 dpf, corresponding to the time when ablations were performed in our manuscript. The two MOs were further characterized in J. R. Panizzi et al. (*Nature Genetics* 2012). Yu et al. (*Nature Genetics* 2008) used an almost identical ATG-blocking MO to determine the phenotypic changes caused by depletion of *foxj1a*, while Ribeiro et al. (*Open Biology* 2017) confirmed the phenotypic changes caused by the ATG-blocking *foxj1a* MO by combined Cas9/gRNA injections.

Using published *foxj1a/foxj1b* MOs, we report that the combined depletion of *foxj1* was not associated with a repair defect despite a typical ciliopathy phenotype characterized by pronephric cysts (marked by stars in Suppl. Fig. 3d). The experiments in Suppl. Figures 2-4 highlight that the migration-mediated repair after a laser-induced injury represents an extremely robust program that is not easily deterred. Neither disruption of canonical or non-canonical Wnt signaling nor interference with apical-basal polarity, ciliogenesis, glomerular filtration and intraluminal flow prevented the repair program. The *foxj1* experiment (Suppl. Figure 3d) represents a nice additional (negative) example, but could easily be omitted from the manuscript without changing any of the conclusions of the paper.

Copious data have been published on *foxj1* MOs; it's unclear why the reviewer feels that they are not well characterized.

Since the reviewer may have confused the *foxj1* MO and *myca* MO data, we have generated a CRISPR/Cas9-based *myca* mutant with a 20 bp-deletion within the 5' region of *myca*. We have now included a series of three independent experiments using this *myca* mutant zebrafish line. To avoid transcriptional compensation, we crossed heterozygote zebrafish for each experiment, and genotyped the fish after performing laser-induced injuries. While 25% of homozygous mutant embryos did not repair, all heterozygote and wild-type embryos repaired (Fischer's exact test, $p = 0.002$). Although the MO-induced phenotype displayed a slightly stronger repair defect ($\approx 40\%$), maternal contribution in the mutant fish can easily explain the difference.

Point 6 –

The authors do not present the expression pattern of *cxcl12a* in the relevant tissue and at the right time as requested. The reference they cite shows embryos at an earlier stage than the time they analyze the process. Importantly, based on the findings of Venkiteswaran 2013 (Fig1C <https://doi.org/10.1016/j.cell.2013.09.046>) *cxcl12a* RNA is actually not expressed in the region of the pronephric duct in 36hpf zebrafish embryos.

The published findings contradict the reviewer remarks, which is unacceptable. The left panel depicts the Figure 1C from the article by G. Venkiteswaran et al., *Cell* 2013, cited by the reviewer. A higher magnification of the pronephros region (middle panel) reveals faint staining for *cxcl12a/sdf1a* at **36 hpf**. The right panel demonstrates clear expression of *cxcl12a* at **48 hpf** (= 2 dpf) (Figure 1C from the cited article by N.B. David, *PNAS* 2002). Thus, *cxcl12a* is expressed in the relevant tissue and at the relevant time point (2 dpf), when injury experiments were performed.

Point 17 –

The criticism relates to the option that the defects in regeneration observed in the *cxcl12a* / *cxcr4b* mutants result from defects related to earlier function of the proteins rather than from a direct role in the process.

Here the authors should have provided *Cxcr4b* until the time of the experiment and should have stopped it just before injury induction (e.g. around 32 hpf). This was not done and the ability to “confuse” the process by *Cxcl12a* expression does not prove that the chemokine functions in the process in a positive direction.

The reviewer ignores the fact that we carefully compared migration speed in wild-type and *cxcl12a/cxcr4b* mutant zebrafish at different time points post fertilization, including the relevant time interval used for laser-induced injuries. This very detailed and laborious analysis revealed no differences between wild-type and mutant fish, arguing strongly against a developmental defect before the injury.

The reviewer's suggestion to "stop Cxcr4b right before the injury" is unrealistic. Although inducible zebrafish lines might be feasible, these tools are currently not available. More importantly, the reviewer completely ignores the fact that the requested experiment was performed in mice with an inducible knockout of *Cxcl12*. This mammalian model is still the "gold standard". It is neither sensible nor necessary to perform the same experiment in two different animal models. Ethic commissions are generally highly averse to approving a duplication of animal experiments.

The heat shock-mediated ectopic expression of *cxcl12a* criticized by the reviewer is essentially based on the experimental approach chosen by Donà et al (*Nature* 2013) to demonstrate the importance of the *Cxcl12a/Cxcr4b* module in LLP migration. Ectopic expression of *Cxcl12* disrupts the repair response, underling the importance of *cxcl12a* and *cxcr4b* to orchestrate the repair in response to a laser-induced injury. Again, the essential role of *Cxcl12/Cxcr4* signaling was validated in a tetracycline-inducible mouse model.

Point 21 –

The answer of the authors concerning the question "CXCR4 signaling is up-regulated to compensate for the loss of *Cxcl12* – how do the authors envision a compensation for lack of ligand by up-regulation of the receptor" is that "The PAX8-driven excision of CXCL12 eliminates CXCL12 from tubular epithelial cells, but does not prevent CXCL12 expression in surrounding tissues or infiltrating cells.". This answer is not supported by any data concerning the expression of *Cxcl12* in neighboring tissues.

We agree with the reviewer that the underlying mechanisms of *Cxcr4* up-regulation are not sufficiently examined. Since this finding is merely observational, we have removed this data from the supplementary material.

Reviewer #2 (Remarks to the Author):

I did not find the response to my comments convincing. "I felt that the setting up of zebrafish pronephros and mouse ischemia reperfusion injury was a contrived system and each part by itself had significant issues though none of them technical. So I am not in favor of publishing this manuscript and I think even if taken at face value, the findings are of minor significance."

These reviewer comments are unprofessional and unacceptable, rather a way to dismiss the paper without fully reading or understanding the manuscript. Zebrafish were used as a screening tool. Conditional mouse knockout models for *Cxcl12* and *Myc* provided critically important observations. A more clinically inclined reviewer would realize that this manuscript has significant implications beyond the field of "acute kidney injury". Our manuscript reveals that the first repair steps after injury are driven by glycolysis, which is hampered by either knockout of *Cxcl12* or *Myc*. Depriving tubular epithelia cells of glucose (for example, by SGLT2 inhibitors) may render patients susceptible to acute kidney injury in the setting of sepsis or other serious disease that affect renal oxygen supply. The FDA has issued a warning that SGLT2 inhibitors may precipitate acute kidney injury. Our data suggests that glucose deprivation (and not the drug itself) is responsible for the increased incidence of acute kidney injury in patients on SGLT2 inhibitors.

Control of glucose concentrations in the setting of sepsis and hypo-perfusion remains a highly controversial topic. Our data suggest that excess insulin and tight glucose control may not be beneficial for organ recovery after ischemia/reperfusion injury. Furthermore, glucose-containing solutions may be preferable to facilitate organ preservation (for example, in organ donation). Finally, our data provide new insight why CXCR4 and MYC facilitate tumor

metastasis: both signaling pathways promote fast cell migration and tumor invasion by augmenting the Warburg effect and facilitating glycolysis. Thus, stating that our findings “taken at face value, are of minor significance” is superficial. We ask the reviewer to more carefully evaluate our manuscript.

It was suggested examining additional connections between the mouse and zebrafish model. We now demonstrate that fast migrating cells such as the lateral line primordium require glycolysis to maintain a migration speed of $\approx 60 \mu\text{m/h}$. In contrast, defective glycolysis has no effect on the slow collective cell migration of the pronephros ($\approx 8 \mu\text{m/h}$). However, blocking glycolysis significantly delayed the repair of a laser-induced injury, revealing that cells with increased migration speed during injury repair depend on glycolysis for additional energy.

We believe that our findings provide novel insight into the function of *Cxcl12/Cxcr4* and *Myc* that could potentially result in new approaches to facilitate the recovery from ischemia/reperfusion injury. Furthermore, our observations provide a rationale why *Cxcr4/Myc*-expressing tumors display a more invasive behavior and a worse prognosis.

Reviewers' Comments:

Reviewer #3:

Remarks to the Author:

I have restricted my review to a couple of technical points including the use of the morpholino which was employed in this study.

The authors apparently use a morpholino that was introduced into the field by Hellman in 2011. I could not find the information in the present paper which of the two MOs published in the Hellman study was used here. Looking back at the 2011 work, the morpholinos were not validated according to current standards: information was provided that the morpholinos have the desired molecular effect on the target mRNA, but this is usually not the problem in MO studies - the issue is whether one can demonstrate that there are no toxic and unspecific effects, and this is something which is entirely missing in the PNAS paper from 2011. The later Nature Genetics paper from the same group just refers to the earlier PNAS work, and does not provide further validation. The authors of the present work state that 'copious studies have used the MOs since', but using the same poorly validated MO in three or four different papers is hardly a sign of confirmation that the MO works without side effects. The only paper which provides additional QCying data is the one by the Saude lab: here, (1) an mRNA lacking the MO-binding site is able to rescue the morphant phenotype in the neural tube, and (2) crispant embryos (i.e. embryos injected with sgRNAs/Cas9 directed against *foxcj1a*) were shown to show similar phenotypes than the morphants. Hence, this is the one paper which attempts to validate the morpholino in question, but the MO is used in a different phenotypic context, and the other morpholino remains completely unvalidated.

Have the authors considered to contact the Saude lab to ask whether mutants were generated and whether these mutants are available? This would be the cleanest way to bring the data quality to a level appropriate for Nature Communications. The way it stands, (1) there is only one MO used, (2) the morpholino used is only partially validated, and validated in a different context, and (3) mutant data are missing. To be clear, this does not fulfil the recently published quality criteria (Stainier et al., 2017).

Concerning the point of gene expression, the authors refer to a PNAS paper by the David group to support expression at 48hpf, but the respective figure (Figure 1C in the right panel) shows expression at 24hpf. Both the shape of the embryo and the figure legend are clear testimonials to this. Hence, the initial reviewer was right to point this out. The authors also point to weak expression in a different paper, which could be taken as evidence for expression in the pronephros, but this data set is not entirely convincing. Frankly, why did the authors not settle this by carrying out a couple of in situ stainings themselves? This would do away with all issues of cross-correlating to other peoples' work, and provide data within the manuscript.

Minor note: the enlarged Fig 4h' is not an enlargement of the region depicted in 4h.

Reviewer #4:

Remarks to the Author:

In this manuscript, Yakulov et al. investigate the migratory repair response of zebrafish pronephros after injury. Using laser ablation, they establish that repair fails when the ablation occurs at 1 dpf but is successful at 2 dpf. In the early timepoint, the injury response results in occlusion through an apparent pursestring closure in contrast with the later timepoint when rapid migratory repair re-establishes a contiguous tubule lumen. After excluding roles for cilia and Wnt signaling in the repair response, the authors performed transcriptome analyses of 1 vs 2 dpf after injury. This identified candidates *Cxcl12a* and *Myca*. Several complementary strategies were used to validate a functional role for these genes and pathways in coordinating the repair response.

Moving to mouse models, the authors created inducible *Cxcl12* and *Myc* tubule deletion mice and

show that deletion of either enhances injury and retards repair. Transcriptomic analysis implicated changes in RA signaling and glycolysis regulated by Cxcl12 and Myc. Exogenous RA rescued the phenotype in Myc knockout mice and pharmacologic inhibition of glycolysis also slowed repair. Together, the authors conclude that Cxcl12 and Myc regulate epithelial repair by inducing glycolysis which allows fast migration.

These are in general well performed studies of a mechanistic and novel nature. Overall I find the conclusions mostly supported by the data. Perhaps the weakest part is the effects of modulation of glycolysis on repair. If the authors could provide a more direct link between Cxcl12 and/or Myc and regulation of glycolysis – for example through overexpression or knockdown in an in vitro model, this would complement the pharmacologic studies in Figure 7.

Reviewer #5:

Remarks to the Author:

Since I am a new reviewer joining the review process at a late stage, my comments are based on the careful evaluation of the latest version of the paper (which I understand it went through multiple revisions), as well as the authors' latest responses to the reviewers. As I see it, the authors have satisfactorily and convincingly addressed several previous technical concerns regarding the use of zebrafish techniques and model, by making multiple and extensive revisions that adequately addressed previous criticism. The latest revisions focused mainly on the metabolic link of fast cell migration during kidney repair.

In general, this paper addresses novel mechanistic details of tissue repair using renal tubule injury as a model that has very broad biological and clinical relevance and importance. The work uses state-of-the-art techniques, zebrafish to establish the basic tubule repair phenomenon using intravital imaging and as a screening tool for subsequent molecular studies. The importance of top molecular candidates is then validated in mouse models, and put back in the zebrafish for final pharmacological work and confirmation.

The major finding is the new mechanistic link and causative role of Cxcl12a/Cxcr4b and Myc signaling-mediated alterations in retinoic acid, mitochondrial metabolism, glycolysis promoting fast cell migration for efficient tissue repair. The results are clear, the main findings are confirmed by multiple experimental approaches, and the conclusions are supported by the data. The findings are consistent with the recently emerging role and importance of altered cell metabolism in both tissue repair and chronic diseases. Importantly, the present work identified new molecular and tissue repair mechanisms that can be targeted in future therapeutic development for not only acute kidney injury, but also for other organ pathologies and cancer.

Reviewers' comments:

Reviewer #3 (Remarks to the Author):

I have restricted my review to a couple of technical points including the use of the morpholino which was employed in this study.

The authors apparently use a morpholino that was introduced into the field by Hellman in 2011. I could not find the information in the present paper which of the two MOs published in the Hellman study was used here. Looking back at the 2011 work, the morpholinos were not validated according to current standards: information was provided that the morpholinos have the desired molecular effect on the target mRNA, but this is usually not the problem in MO studies - the issue is whether one can demonstrate that there are no toxic and unspecific effects, and this is something which is entirely missing in the PNAS paper from 2011. The later Nature Genetics paper from the same group just refers to the earlier PNAS work, and does not provide further validation. The authors of the present work state that 'copious studies have used the MOs since', but using the same poorly validated MO in three or four different papers is hardly a sign of confirmation that the MO works without side effects. The only paper which provides additional QCing data is the one by the Saude lab: here, (1) an mRNA lacking the MO-binding site is able to rescue the morphant phenotype in the neural tube, and (2) crispant embryos (i.e. embryos injected with sgRNAs/Cas9 directed against *foxj1a*) were shown to show similar phenotypes than the morphants. Hence, this is the one paper which attempts to validate the morpholino in question, but the MO is used in a different phenotypic context, and the other morpholino remains completely unvalidated.

Have the authors considered to contact the Saude lab to ask whether mutants were generated and whether these mutants are available? This would be the cleanest way to bring the data quality to a level appropriate for Nature Communications. The way it stands, (1) there is only one MO used, (2) the morpholino used is only partially validated, and validated in a different context, and (3) mutant data are missing. To be clear, this does not fulfil the recently published quality criteria (Stainier et al., 2017).

We have removed the *foxj1* MO data. Although importing sperm from the *foxj1* mutant fish is an option, it will likely take an additional 3+ months to raise adult fish required to perform the laser-ablation in embryos. As already pointed out before: the *foxj1* MO data is dispensable, and represents another example to demonstrate the robustness of the repair response. In addition, validating a negative readout (i.e. the lack of a repair defect) in a mutant zebrafish line is problematic; it would be virtually impossible to differentiate between a negative result and genetic compensation.

Concerning the point of gene expression, the authors refer to a PNAS paper by the David group to support expression at 48hpf, but the respective figure (Figure 1C in the right panel) shows expression at 24hpf. Both the shape of the embryo and the figure legend are clear testimonials to this. Hence, the initial reviewer was right to point this out. The authors also point to weak expression in a different paper, which could be taken as evidence for expression in the pronephros, but this data set is not entirely convincing. Frankly, why did the authors not settle this by carrying out a couple of in situ stainings themselves? This would do away with all issues of cross-correlating to other peoples' work, and provide data within the manuscript.

We confirmed the previously reported expression of *Cxcl12a* at the relevant time point by *in situ* hybridization (Supp. Fig. 6 g).

Minor note: the enlarged Fig 4h' is not an enlargement of the region depicted in 4h.

This was corrected.

--

Reviewer #4 (Remarks to the Author):

In this manuscript, Yakulov et al. investigate the migratory repair response of zebrafish pronephros after injury. Using laser ablation, they establish that repair fails when the ablation occurs at 1 dpf but is successful at 2 dpf. In the early timepoint, the injury response results in occlusion through an apparent pursestring closure in contrast with the later timepoint when rapid migratory repair re-establishes a contiguous tubule lumen. After excluding roles for cilia and Wnt signaling in the repair response, the authors performed transcriptome analyses of 1 vs 2 dpf after injury. This identified candidates *Cxcl12a* and *Myc*. Several complementary strategies were used to validate a functional role for these genes and pathways in coordinating the repair response.

Moving to mouse models, the authors created inducible *Cxcl12* and *Myc* tubule deletion mice and show that deletion of either enhances injury and retards repair. Transcriptomic analysis implicated changes in RA signaling and glycolysis regulated by *Cxcl12* and *Myc*. Exogenous RA rescued the phenotype in *Myc* knockout mice and pharmacologic inhibition of glycolysis also slowed repair. Together, the authors conclude that *Cxcl12* and *Myc* regulate epithelial repair by inducing glycolysis which allows fast migration.

These are in general well performed studies of a mechanistic and novel nature. Overall I find the conclusions mostly supported by the data. Perhaps the weakest part is the effects of modulation of glycolysis on repair.

If the authors could provide a more direct link between *Cxcl12* and/or *Myc* and regulation of glycolysis – for example through overexpression or knockdown in an *in vitro* model, this would complement the pharmacologic studies in Figure 7.

We found that over-expression of *Cxcl12* or *Myc* in tumor cells (HEK 293T/ HeLa cells) as suggested had no effect. We believe that cultured tumor cells have adapted to display a maximal Warburg effect/glycolytic flux, and that it is difficult to exert an additional increase in glycolytic capacity.

We therefore decided to isolate primary renal tubular epithelial cells from wild-type and *Myc* KO mice, induced the excision of *Myc in vitro*, and measured the glycolytic capacity. These experiments (Suppl. Fig. 23) clearly revealed that the glycolytic flux is impaired in *Myc*-deficient cells, supporting our hypothesis.

--

Reviewer #5 (Remarks to the Author):

Since I am a new reviewer joining the review process at a late stage, my comments are based on the careful evaluation of the latest version of the paper (which I understand it went through multiple revisions), as well as the authors' latest responses to the reviewers. As I see it, the authors have satisfactorily and convincingly addressed several previous technical concerns regarding the use of zebrafish techniques and model, by making multiple and extensive revisions that adequately addressed previous criticism. The latest revisions focused mainly on the metabolic link of fast cell migration during kidney repair.

In general, this paper addresses novel mechanistic details of tissue repair using renal tubule injury as a model that has very broad biological and clinical relevance and importance. The work uses state-of-the-art techniques, zebrafish to establish the basic tubule repair phenomenon using intravital imaging and as a screening tool for subsequent molecular studies. The importance of top molecular candidates is then validated in mouse models, and put back in the zebrafish for final pharmacological work and confirmation.

The major finding is the new mechanistic link and causative role of Cxcl12a/Cxcr4b and Myc signaling-mediated alterations in retinoic acid, mitochondrial metabolism, glycolysis promoting fast cell migration for efficient tissue repair. The results are clear, the main findings are confirmed by multiple experimental approaches, and the conclusions are supported by the data. The findings are consistent with the recently emerging role and importance of altered cell metabolism in both tissue repair and chronic diseases. Importantly, the present work identified new molecular and tissue repair mechanisms that can be targeted in future therapeutic development for not only acute kidney injury, but also for other organ pathologies and cancer.

Reviewers' Comments:

Reviewer #3:

Remarks to the Author:

The authors have addressed my comments. In particular, morpholino data which were insufficiently validated were removed, and the ISH has now been provided in suppl. Figure 6.

I have no further comments.

Reviewer #4:

Remarks to the Author:

The authors have satisfied my concern.